# A Review of the Current State of Magnetic Force Microscopy to Unravel the Magnetic Properties of Nanomaterials Applied in Biological Systems and Future Directions for Quantum Technologies

**DOI:** 10.3390/nano13182585

**Published:** 2023-09-18

**Authors:** Robert Winkler, Miguel Ciria, Margaret Ahmad, Harald Plank, Carlos Marcuello

**Affiliations:** 1Christian Doppler Laboratory—DEFINE, Graz University of Technology, 8010 Graz, Austria; robert.winkler@felmi-zfe.at (R.W.); harald.plank@felmi-zfe.at (H.P.); 2Instituto de Nanociencia y Materiales de Aragón (INMA), CSIC-Universidad de Zaragoza, 50009 Zaragoza, Spain; ciria@unizar.es; 3Departamento de Física de la Materia Condensada, Universidad de Zaragoza, 50009 Zaragoza, Spain; 4Photobiology Research Group, IBPS, UMR8256 CNRS, Sorbonne Université, 75005 Paris, France; margaret.ahmad@sorbonne-universite.fr; 5Graz Centre for Electron Microscopy, 8010 Graz, Austria; 6Institute of Electron Microscopy, Graz University of Technology, 8010 Graz, Austria; 7Laboratorio de Microscopias Avanzadas (LMA), Universidad de Zaragoza, 50018 Zaragoza, Spain

**Keywords:** atomic force microscopy, biological systems, drug delivery, magnetic force microscopy, magnetic properties, magnetic tip fabrication, nanofabrication, quantum technologies, single-molecule studies

## Abstract

Magnetism plays a pivotal role in many biological systems. However, the intensity of the magnetic forces exerted between magnetic bodies is usually low, which demands the development of ultra-sensitivity tools for proper sensing. In this framework, magnetic force microscopy (MFM) offers excellent lateral resolution and the possibility of conducting single-molecule studies like other single-probe microscopy (SPM) techniques. This comprehensive review attempts to describe the paramount importance of magnetic forces for biological applications by highlighting MFM’s main advantages but also intrinsic limitations. While the working principles are described in depth, the article also focuses on novel micro- and nanofabrication procedures for MFM tips, which enhance the magnetic response signal of tested biomaterials compared to commercial nanoprobes. This work also depicts some relevant examples where MFM can quantitatively assess the magnetic performance of nanomaterials involved in biological systems, including magnetotactic bacteria, cryptochrome flavoproteins, and magnetic nanoparticles that can interact with animal tissues. Additionally, the most promising perspectives in this field are highlighted to make the reader aware of upcoming challenges when aiming toward quantum technologies.

## 1. Introduction

The properties of biological systems are determined by the ubiquitous forces that govern in nature and define their performance. There exist four fundamental forces named strong nuclear, electromagnetic, weak nuclear, and gravitational, which are classified from the strongest interactions to the weakest. Magnetism is not typically considered a significant factor in biological systems. For this reason, many existing mysteries surrounding its role in biology are currently not well understood. Many examples of external, either static or oscillating, magnetic fields (*B*) that affect biological systems have been reported, such as the cases of cryptochromes [1], magnetosomes from magnetotactic bacteria of different strains [2], regulation of the calcium concentration as a cell viability key factor [3], and enzymatic reactions related to DNA synthesis [4], among others. Cryptochromes are proteins involved in light signaling or in circadian rhythms in many plants and animals and appear to sense *B* [5]. Magnetosomes are magnetic nanoparticles coming from a natural source like magnetotactic bacteria and can be a suitable alternative to chemically synthesized nanoparticles designed for hyperthermia treatments [6]. Other biological reactions that take place inside the cell are also affected by *B*. Magnetic control of biochemistry and catalytic functions of biomolecules require a better comprehension of this property and of how it impacts biological systems.

There are several bulk techniques available to measure the magnetic properties of the studied samples.

Superconducting quantum interference device (SQUID) magnetometry is a highly sensitive technique to measure ultralow magnetic signals up to 5 × 10^−14^ Tesla (T) with a noise threshold of nearly 3 fT · Hz^−1/2^ [7]. The signal-to-noise ratio can be improved when the SQUID is damped by low susceptometer resonances [8]. For this reason, SQUID is considered the most sensitive type of quantitative magnetometry with 1 · 10^−8^ electromagnetic units (emu; 1 emu = 10^−3^ Am^2^). SQUID consists of placing a sample in a magnetic field and measuring the magnetic moment of the sample as a function of the applied field strength [9]. Moreover, controlling the applied current by the integration of a heating resistor on the same sample chip makes tunable the SQUID sensor device [10]. This technology has been applied to measure the magnetic properties of biological systems such as magnetosomes from magnetotactic bacteria [11], mesenchymal stem cells for tissue engineering applications [12], or the characterization of iron oxide nanoparticles in biological samples [13], and for the magnetic separation of microplastic bodies from water resources [14], respectively.Vibrating sample magnetometry (VSM) consists of the sample mounting on a thin rod under *B* while simultaneous vibration of the sample at a specific frequency occurs [15]. Multiple magnetic properties including magnetic moment, susceptibility, and coercivity can be measured by varying the strength of *B*, the sample orientation with respect to *B*, and its temperature [16]. Nowadays, customized VSM setups can achieve signal sensitivities ranging from 1 · 10^−5^ to 1 · 10^−6^ emu [17]. The employment of VSM has revealed the metagenomic analysis of magnetotactic bacteria [18], the detection of ferromagnetic materials in insect tissues responsible for their orientation toward external magnetic fields under both light and dark conditions [19], and the characterization of magnetic nanoparticles in the use of DNA isolation [20] or for hyperthermia therapies [21].Magneto-optic Kerr effect (MOKE) is based on measuring the rotation of the reflected light polarization, which is proportional to the magnetic moment of the sample under an external magnetic field [22]. MOKE technology can be used to address many magnetic sample properties like the magnetization process, the magnetic domain structure, and the magnetic anisotropy [23]. The sensitivity of MOKE is slightly higher than VSM, being settled at 1 · 10^−7^ emu [24]. Recently, a twofold increase in the intensity signal was reported by polarizing the beam splitter-based MOKE setup [25] and the reach of femtosecond-scale time resolution by coupling a free electron laser [26], respectively. The assemblies of magnetosomes from magnetotactic bacteria [27], and the manipulation and trap of magnetic yeast cells on lab-on-chip devices [28] are some of the few examples where MOKE is employed in biological samples. MOKE is conventionally used in the study of multiferroic materials [29].Magnetic resonance techniques like nuclear magnetic resonance (NMR) [30], electron paramagnetic resonance (EPR) [31], or ferromagnetic resonance (FMR) [32] apply a magnetic field to the sample and measure the subsequent response of the atomic or electronic spins according to this field, respectively. While NMR is mainly focused on the determination of molecular structure and dynamics independent of the nature of the sample [33], EPR requires magnetic domains embedded inside the sample structure. For this reason, NMR cannot be considered a bulk technique to measure the magnetic properties of biosystems. On the other hand, EPR is capable of detecting 10^12^ spins per mT linewidth [34]. Alternatively, FMR detects the magnetic moments of non-paramagnetic materials by applying a second microwave pulse, being widely used for ferromagnetic particles [35] or magnetosomes [36]. EPR has satisfactorily ascertained the magnetic sensitivity of cryptochromes in birds [37], the catalytic mechanisms of molecular radicals existing in nature [38], the conformational dynamics of membrane proteins [39] or metalloenzymes [40], and the electron spin relaxation of porphyrins [41], which regulates oxygen transport in the blood and muscles.Mössbauer spectroscopy is a versatile technique to study the interaction of certain isotopes with their surroundings [42]. This technology enables the hyperfine interactions between the nuclei and electrons to be measured with an accuracy of 14–15 magnitude orders [43]. Mössbauer spectroscopy was used to unravel the magnetic properties of nanometer-size particles [44], magnetosomes [45], and cryptochromes [46]. Mössbauer spectroscopy is a particularly powerful instrument when it is exploited in combination with synchrotron facilities [47].Alternating current (AC) susceptibility refers to the extent to which the material can become magnetized in response to an alternating magnetic field [48]. AC susceptibility measurements are commonly carried out through a magnetic susceptibility meter or a vibrating sample magnetometer. The signal analysis for the moment measurements is processed by a high-speed digital voltmeter leading to sensitivity yields of nearly 3.5 × 10^−6^ emu [49]. AC susceptibility measurements allow for the detection of directional changes in magnetic fields of small insects [50] and magnetotactic bacteria [51], the nucleus positioning of carcinogenic cells [52], iron detection in ferritins or hemoglobins [53] and human serum albumin [54], or the elasticity of globular proteins labeled with gold nanoparticles [55].

The main limitations of the above-described bulk techniques are based on the lack of sensitivity to detect singularities or the hidden phenomena at very specific sample local areas. These detrimental aspects do not allow the detection of single events, which are crucial to gaining the underpinning knowledge of magnetic biological nanomaterials. The high complexity of living systems hinders the achievement and subsequent full comprehension of these magnetic properties [56]. For all these reasons, single-molecule techniques have emerged as suitable alternatives to overcome these drawbacks. 

Firstly, scanning transmission electron microscopy (STEM)-based techniques are encompassed by Lorentz transmission electron microscopy (LTEM), holography, and differential phase contrast (DPC) microscopy that are capable of measuring magnetic signals. Typically, TEM measurements are devoted to visualizing the supramolecular architecture of materials including biological specimens [57]. LTEM records the interaction between the primary electrons and the magnetic fields, which emerged in the tested sample [58]. The electron deflection suffered when they encounter sample regions with different magnetic domains is used to create the magnetic contrast image. The main limitations of LTEM are the sensitivity to beam conditions, such as the incidence angle, the electron acceleration energy, and the phase retrieval process, which exist when the electron wave phase information is extracted. Thus, the implementation of these numerical simulations in combination with the whole process of digital content acquisition involving the proper generation, coding, and quality assessment of the digital holograms is time-consuming in comparison to other techniques where no holograms are requested to be recorded. Finally, STEM DPC, a segmented detector, measures the deflection of electrons caused by the local magnetic field of the specimen due to Lorentz’s force. The position of the deflected electrons on the four-quadrant annular detector can be translated to the in-plane direction of the magnetic field at each point of the scanned area. DPC provides images, gathering the change in the phase caused by the changes in sample thickness, refractive index, or density [59]. Eventually, a phase plate optical device is inserted into the STEM setup to expand the boundaries of classical STEM by introducing a controlled phase shift to the electron wavefronts and converting it to an amplitude [60], enabling the possibility of interrogating surfaces with weak phase contrasts. On the other hand, DPC shows some challenges like the potential artifact sources coming from the proper calibration and alignment of the phase plate, or the beam tilt. LTEM [61], holography [62], and DPC [63] have been successfully employed to unravel the magnetic properties of a broad panoply of surface materials. Nevertheless, the following limitations need to be considered in STEM-based techniques. The samples for STEM measurements must be electron transparent, which limits these techniques to very thin samples and often involves complicated sample preparation procedures. Furthermore, the above-described STEM-based techniques can cause structural damage on the sample surface, particularly at high beam currents. This aspect considerably limits their application in those radiation-sensitive materials with a special focus on soft matter and biological systems. Promising results have been found working at low-density currents of 0.1 e^−^Å^−2^s^−1^ at room temperature conditions [64] or integrating different phase contrasts [65], but much effort still needs to be devoted to this field. 

Then, optical [66] and magnetic [67] tweezers (OT and MT, respectively) are tools that enable the force measurement toward the positioning of the sample at the micrometer scale. The noise threshold of both of them is very low. Additionally, MT can interfere with the magnetic properties of certain materials or biological systems. Recently, MT devices were designed to integrate the real-time feedback control of the magnetic flux density by using a proportional–integral–derivative (PID) controller and a cascade control scheme [68]. The optimization of PID gains by the implemented algorithms results in magnetization response times below 100 ms, which significantly minimizes the negative interferences with magnetic samples. Nevertheless, as a counterpart, the main limitation of OT and MT is their inability to map the sample and simultaneously correlate the property of interest with the topography of the tested features. This makes OT and MT unappealing for single-molecule studies on magnetic biological systems.

Single-probe microscopies (SPMs) were shown to be a suitable alternative to overcome the aforementioned drawbacks existing in OT and MT. Since atomic force microscopy (AFM) was discovered in 1986 [69], a large volume of research has been devoted to coping with the challenges showcased by biological systems. AFM is considered a multiparametric technique [70] capable of addressing a multitude of physico-chemical properties of the tested biological samples like the morphology of biomolecular complexes [71] or living cell scaffolds [72]. Three-dimensional volumetric statistical studies of the observed biomolecular morphology features upon the presence of their ligands and cofactors can be carried out by subtracting the background contribution through accurate mask thresholds [73]. This information allows a better understanding of some relevant processes driven in biology like the conformational transitions taking place in flavoenzymes under certain redox conditions [74], the structural and functional properties of red blood cells [75] exhibited in health and disease [76], the topological changes in lignocellulosic biopolymers after chemical grafting [77], which can be exploited for food packaging or in the design of green-friendly composites [78], or the creation of novel materials for implantable tissue-engineered devices [79]. Then, the intermolecular adhesion properties of biological systems can be addressed by AFM-based force spectroscopy (AFM-FS) studies [80]. This operational mode allows the determination of the rupture force events between functionalized tips and chemically modified surfaces. AFM-FS has been successfully employed to discern the range of intermolecular interactions between redox enzymes with their protein partners [81] or coenzymes [82], recognition events between biomolecules [83] or cells [84], bioluminescence-engineered proteins [85], or the adhesion displayed between natural plant fibers and biopolymer matrices [86]. The nanomechanical properties of biological soft matter can be deciphered by nanoindentation studies [87]. The AFM tip works as an indenter to exert an elastic deformation on the tested surface sample. Nanoindentation was employed to measure the elastic properties of living cells [88], collagen fibrils [89], and biopolymers under different relative humidity [90], and to monitor the decay of cellular nanomechanics in diseases like cystic fibrosis [91] or cancer [92], together with drug-targeting effectiveness [93]. Finally, AFM based on nanoscale infrared spectroscopy (AFM-nanoIR) enables the mapping of chemical composition and topography or determination of the full IR spectra at specific sample locations [94]. AFM-nanoIR works by scanning the sample with a conventional AFM tip at the same sample location where a pulsed tunable IR beam spot is focused, causing a photothermal expansion, which is recorded and subsequently correlated with the chemistry of the analyzed area. The monitoring of molecular aggregation inhibitors to neurodegenerative amyloidogenic proteins [95], the extracellular vesicle formation [96], or the underlying drug release mechanisms relying on their content load [97] are some examples of AFM-nanoIR studies.

Magnetic force microscopy (MFM) is a non-contact force tool where the magnetic domains are imaged by the interactions exerted between the magnetized AFM tip and the external sample surface areas [98]. MFM has become an appealing technique to detect magnetic forces at the single molecule level based on the extremely high resolution compared to other magnetic bulk techniques such as MOKE where the spatial resolution is limited to the optical wavelength (approximately 500–600 nm) [99]. Even SQUID is far below the stray field of single atomic magnetic layers [100]. The lateral resolution of MFM relies on the coated AFM tip radius of the commercial AFM probes (30–60 nm). MFM combines the ultrahigh-sensitivity detection of magnetic forces with the capability of imaging and positioning at the nanometer scale. Additionally, MFM allows environmental measurements, unlike other techniques such as EPR where cryogenic temperatures are required to enhance the setup sensitivity [101]. Other advantages showcased by MFM are that soft matter samples can be interrogated, which is not recommended in other techniques such as VSM where fragile materials can be damaged by the magnetometer vibrations during the scan. Finally, MFM presents the possibility of measuring in liquid media [102], mimicking inner cellular conditions of the studied biology systems in contrast to all the rest of the bulk magnetic techniques or conducting experiments at ultralow temperatures [103]. By coupling the MFM setup with a high-vacuum cryostat containing cryogenic liquids such as nitrogen or helium, the scanning temperature can reach 64–70 K or 4 K, respectively [104]. This configuration allows the interrogation of fast magnetic relaxation changes unable to be detected under environmental conditions [105]. For all these aforementioned reasons, MFM successfully copes with the pitfalls existing in bulk magnetometric measurements.

This review aims to highlight the enormous potential of MFM to reveal the biological processes where magnetism is involved, based on the recent technological developments that this technique has undergone. The manuscript is divided into the following sections in order to make the information more comprehensible: (1) Introduction, (2) Biological systems affected by magnetism, (3) Working principles of MFM, (4) Existing MFM operational modes, (5) Previous magnetic force measurements with commercially available MFM tips, (6) Development of ultra-sharp MFM tips, and (7) Discussion and future perspectives. We expect to sensitize the readers to the promising avenues that MFM can open up in biology, and which could be implemented in many industrial applications like drug delivery or quantum technologies.

Even if the impact of magnetism in biology is not as evident as other environmental physico-chemical factors like temperature, relative humidity, pH, ionic strength, oxygen availability, and mechanical forces, among others, there exist many examples of how magnetism intersects with biology. The next section aims to list the existing biological systems affected by magnetism and provide the necessary details to the reader for gaining particular knowledge of the pivotal role that the magnetic forces exert on them and the potential applications arising in this field.

## 2. Biological Systems Affected by Magnetism

### 2.1. Magnetosomes from Magnetotactic Bacteria

Magnetotactic bacteria (MTB) have the ability to produce intracellular membrane-bound organelles containing ferrimagnetic nanoparticles called magnetosomes [106] Magnetosomes are composed of magnetite (Fe_3_O_4_) or greigite (Fe_3_S_4_) and under anoxic conditions MTB can biomineralize and arrange chains up to 40 magnetosomes at mid cell (Figure 1a) [107]. This provides MTB the possibility to sense *B* and move toward them [108]. This phenomenon is named magnetotaxis. The mechanism to form magnetosomes is divided into four different stages (Figure 1b). 

The first step is based on protein sorting and invagination. MTB recruit magnetosome-associated proteins (MAPS) including MamB, MamL, MamM, or MamQ, which are crucial for membrane formation. Deletion of the genes that produce these proteins causes the formation of immature vesicles unable to biomineralize magnetosomes [109]. Then, the cellular arrangement of magnetosomes is organized by MamK and MamY. MamK is an actin-like ATP-dependent protein that assembles in filaments [110], while MamY detects the highest convex area in the formed vesicle membranes and aids in properly aligning the magnetosome chain [111]. Ion transport inside the vesicle membrane is the third stage that is entailed in the cooperating action of MamB and MamM. These proteins are cation diffusion facilitator transporters that regulate the iron transfer from the cytoplasm to the magnetosome vesicles exploiting the proton motive force [112]. Crystal shape and size control of magnetosomes during nucleation is the last process tightly regulated by genetic and environmental factors. Genes encoding the MamA/B proteins are involved in controlling the crystal morphology, whereas MamC and Mms6 are in charge of the nanoparticle size [113]. Environmental factors like temperature, PH, or oxygen concentration in the media have been shown to affect magnetosome shape and size [114]. Room temperatures favor the development of spherical cuboctahedral crystals [115] while at lower temperatures, the formation of elongated magnetosomes is reported to be induced [116].

Magnetosome nanoparticles can be exploited in a wide range of biomedical and biotechnology applications like the regulation of iron homeostasis in living cells [117], or their use as microrobots for targeted cancer therapies [118] or against microvascular thrombolysis [119]. Recently, genetic engineering modifications of those MTB crucial proteins involved in the biomineralization process triggered the formation of customized magnetosome shapes, like a bullet shape [120]. Due to their significant shape anisotropy, the magnetic coercivity properties of these magnetosomes are highly improved compared to the classic spherical or cuboctahedral, which make them excellent candidates for magnetic resonance imaging (MRI) contrast agent [121] or magnetic hyperthermia treatments [122].

**Figure 1 nanomaterials-13-02585-f001:**
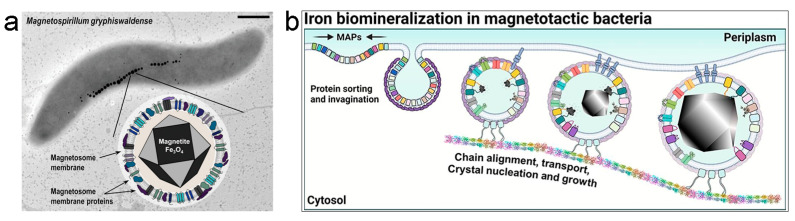
(**a**) Magnetosome formation in *M. gryphiswaldense*. Scale bar 0.5 µm. Reprinted with permission from [123]. Copyright 2023, Wiley. (**b**) Suggested model of protein sorting, membrane invagination, and magnetosome assembly into an organized chain. Reprinted with permission from [124]. Copyright 2021, Elsevier.

### 2.2. Synthetic Magnetic Nanoparticles Used for Biological Applications 

There exist many types of magnetic nanoparticles (MNPs) with different chemical natures designed by synthetic routes. Firstly, cobalt nanoparticles treasure high magnetic anisotropy properties and can be easily magnetized in one direction [125]. For this reason, cobalt nanoparticles can be exploited in applications like energy storage or used as antimicrobial agents based on the observed high antibacterial activity [126]. Then, gadolinium nanoparticles enhance the contrast of magnetic resonance imaging due to their high magnetic moment [127]. Nevertheless, the risk of gadolinium release limits their clinical applications. For this reason, more research needs to be devoted to creating more efficient strategies for the encapsulation and surface labeling of gadolinium nanoparticles [128]. Substitution strategies of ferrite nanoparticles by europium [129] or samarium [130] offer suitable alternatives to gadolinium MNPs. Nickel nanoparticles produced by thermal decomposition are another class of MNPs with promising applications in magnetic data storage in electronic devices owing to their excellent electrical properties [131] or for the production of the next-generation electrodes for supercapacitors [132].

Here, iron oxide (Fe_2_O_3_) nanoparticles are discussed since they are the most commonly synthesized MNPs due to the high availability of iron, their excellent displayed biocompatibility, biodegradability, and overall chemical stability [133]. This aspect makes Fe_2_O_3_ nanoparticles suitable for long-term applications. There exist several methods to produce this type of nanoparticle as co-precipitation, thermal decomposition, or sol–gel. The main limitation found in Fe_2_O_3_ nanoparticles is their tendency to self-aggregate due to a high ratio between the surface area and the MNP volume, in addition to the attractive magnetic interactions among magnetite [134]. To prevent this detrimental effect, many strategies have been developed like surface grafting with surfactants or biodegradable and biocompatible polymers, and the control of the pH, sonication, and MNP size [135]. The coating of Fe_2_O_3_ MNPs with polyethylene glycol (PEG) chains not only serves as a particle stabilizer but also offers the possibility of functionalizing the external PEG side with chemical compounds of interest, making this approach interesting for hyperthermia and smart drug delivery [136]. Smaller Fe_2_O_3_ nanoparticles are more stable in liquid solutions. The shape and morphology of Fe_2_O_3_ nanoparticles are tunable depending on their final applications (Figure 2a–f). Cubic Fe_2_O_3_ MNPs show the optical shape for magnetic stem cell tracking imaging (Figure 2a) [137]. Fe_2_O_3_ MNP nanoflowers encapsulated in liposomes display an outstanding response in hyperthermia treatments against lung adenocarcinoma (Figure 2b) [138]. Hexagonal iron plates (Figure 2c) [139] and rod-shape (Figure 2d) [140] Fe_2_O_3_ nanoparticles display stronger magnetic relaxation times compared to classical globular MNP morphology (Figure 2e) [141], which can be particularly interesting for longer-lasting imaging of biological tissues and monitoring disease progression in response to certain treatments [142]. Finally, iron oxide tetrahedrons (Figure 2f) [139] also display optimal optical properties [143] for ultrasensitive biosensing multimodal MRI and hyperthermia therapies. 

Thus, the iron oxide MNP shape is an important factor that impacts their properties and performance, but the size of the nanoparticle is another key point to consider in their design and synthesis. Figure 2g depicts the logarithm decreasing the magnetic saturation of globular Fe_2_O_3_ MNPs according to their diameters [144]. The nanoparticle magnetization decreases nearly 2.5-fold when the particle diameter dimensions are reduced 5.5 times (from 14.0 to 2.5 nm). The smallest size nanoparticles undergo a weak impact of the dipolar interactions on their magnetic behavior [145]. For this reason, small MNPs show enhanced magnetic properties rather than larger particles. In addition, small MNPs are the best option for targeted delivery or assistance for microfluidic analysis [146] due to their increased surface load areas, and the possibility of embedding these MNPs in hydrogels or liposomes can render smart platforms for tissue wound healing [147]. All the above-described considerations highlight the urgency of developing new characterization methodologies for those MNPs with small dimensions by improving the current limit of sensitivity. 

### 2.3. Enzymatic Reactions Involving DNA and Neurodegenerative Diseases 

DNA replication is the process where living cells produce exact copies of their genetic material and it is divided into the following three stages: initiation, elongation, and termination [148]. DNA replication involves multiple enzymes, which work together in a highly coordinated manner. One of the most relevant chemical reactions is based on phosphorylation [149]. This enzymatic pathway catalyzes the transfer of phosphate (PO_4_^3−^) groups from energetic free adenosine triphosphate (ATP) molecules to DNA strands. Phosphorylation activates the origin recognition complex (ORC) to recruit other proteins and bind the DNA [150]. Then, the DNA polymerase adds nucleotides to the forming DNA strand during the elongation phase through ATP hydrolysis [151]. Finally, phosphorylation also regulates DNA replication by the activation of proteins like the checkpoint kinase 1 (Chk1), which delays the cell cycle progression under replication stress conditions or as a response to DNA damage [152]. Furthermore, phosphorylation is not only crucial to regulate DNA replication but, also, to control the energy supply in the cellular and tissue metabolisms [153], the proper functioning of the immunological system [154], and the muscular contraction [155], among others. Recently, it was discovered that the impact of *B* on the ATP synthesis and subsequent phosphorylation process took place during DNA replication [156]. The rate of ATP synthesis by creatine kinase increased nearly 3.5-fold at *B* of 55 mT when the ^24^Mg nonmagnetic ions were exchanged by the respective paramagnetic ^25^Mg [157]. Sometimes the application of *B* has contradictory effects. For example, its presence has demonstrated the promotion of the DNA double-strand repair process in human bronchial epithelial cells [158]. On the other hand, strong *B* can induce the production of reactive oxygen species (ROS) and the resulting DNA damage in human neuroblastoma cells [159], spermatozoa [160], or cultured mammalian cells [161]. For this reason, it is necessary to carry out more research on this topic by designing customized ultrasensitive detection strategies at the single-molecule level. The gathered knowledge could serve not only to gain more insights according to the sequential binding and location of protein regulators on DNA strands [162], which regulates the organism gene expression [163] or the DNA degradation beyond the proteasome activation [164], but also to optimize those industrial technologies related to DNA, such as clustered regularly interspaced short palindromic repeat-associated protein 9 (CRISPR-Cas9), which enables edit-specific genes in an organism that can be exploited in drug delivery [165] or agricultural biotechnology [166], among others.

Neurodegenerative disorders (NDs) are a group of diseases that cause progressive dysfunction of neurological performance. NDs encompass several types including Alzheimer’s, Parkinson’s, Huntington’s, and Creutzfeldt–Jakob disease or amyotrophic lateral and multiple sclerosis. All these diseases are characterized by the gradual loss of function of nerve cells, which leads to decay in cognitive, sensory, and motor abilities. A total of 10 million deaths and 349 million disability-adjusted life years were reported in 2019 caused by NDs [167]. For this reason, the understanding of the underlying mechanisms and main factors that trigger these disorders are aspects of key importance. Many studies at the molecular level have been carried out including AFM imaging to visualize the morphological folding and unfolding conformations of amyloidogenic fibrillar proteins like tau [168], amyloid beta (Aβ) [169], huntingtin [170], and the S100B family [171], under conditions that mimic the conditions inner the brain. Nevertheless, the biological detection of weak *B* is a topic with numerous remaining open questions. Those reactions involving radical spins are susceptible to being affected by *B*. The increase in ROS and oxidative stress fosters protein misfolding and aggregation and the incidence of NDs [172]. In addition, weak magnetic fields lead to more efficient phosphorylation processes, as above described. When amyloidogenic proteins like tau are abnormally phosphorylated, they cause the formation of fibrillary tangles responsible for neuronal apoptosis [173]. Thus, electromagnetic fields can interfere with the evolution of neurodegenerative diseases [174] like amyotrophic lateral sclerosis [175]. These examples highlight the importance of deeply interrogating the role that magnetic forces play at the biomolecular level inside living cells and what the biological response is to external magnetic stimuli. Thus, this fundamental knowledge could significantly contribute to creating more efficient methodologies and advanced tools to exploit the controlled application of *B* for the early prognosis and the subsequent treatment of human diseases. 

### 2.4. Cryptochromes

The singular mechanism that enables the cryptochromes as a unique biological receptor to sense spin chemical forces requires a full explanation of how this protein family responds to light as well as *β*. The cryptochromes are members of a vast and diverse family of flavoproteins that were initially discovered in plants [176]. Here, they control many aspects of plant growth and development in response to blue light, and have important agricultural significance [177,178,179]. Cryptochromes are very similar to a known family of DNA repair enzymes called photolyases, which are globular flavoproteins that utilize light energy for the repair of UV-damaged DNA, either of (6–4) photoproducts or of cyclobutane pyrimidine dimers [180,181]. Photolyases bind two light-harvesting co-factors: a catalytic flavin adenine dinucleotide (FAD) and a secondary light-harvesting antenna pigment (folate or flavin derivative) [180]. In the case of plant cryptochromes, there is significant homology to so-called microbial Type I photolyases, specifically within the N-terminal flavin binding domain of approximately 500 amino acids (see e.g., Figure 3). Plant cryptochromes were unable to repair DNA, did not bind to a light-harvesting antenna cofactor, and had, in addition, a 200 amino acid C-terminal extension beyond the region of homology to photolyases. Both N-terminal and C-terminal domains participate in interaction with signaling proteins and can undergo conformational change [182,183,184] (Figure 3). However, not all cryptochromes have significant C-terminal extensions, and some of them can still repair DNA. A more general definition that fits most cryptochromes is that they are photolyase-like proteins that have either lost their significant DNA repair function, have gained novel cellular signaling functions, or both. 

Cryptochromes are found throughout the biological kingdom, being present in archaebacteria and many species of prokaryotes and eukaryotes including in humans [185,186]. This is at least in part due to their having independently arisen several times in evolution, apparently from different photolyase ancestors, but with converging roles and mechanisms of action. Much interest has centered around animal and human cryptochromes, which have roles in the circadian clock and are implicated in diseases ranging from diabetes to cancer. Intriguingly, some cryptochromes, such as the Type II mammalian cryptochromes, do not require light for certain signaling roles in the circadian clock [187]. 

Mechanistically, all cryptochromes are capable of undergoing redox reactions, whereby the protein-bound flavin can be reduced by light. Reduced flavin can subsequently be reoxidized in the presence of molecular oxygen. This process, known as ‘photoactivation’, has been conserved from photolyases where it serves to keep the catalytic flavin in the reduced redox state for DNA repair [185,188,189]. The best-studied cryptochromes, which have known light signaling roles, are from plant (Cry1 and Cry2) and drosophila (Type I Cry) [177]. In both these cases, flavin photoreduction forms the basis of a photocycle in which oxidized flavin (FADox) is reduced by light to various reduced (FAD^o−^, FADH^ο^, and FADH^−^) redox states with accompanying intraprotein electron and/or proton transfer events. These redox state transitions activate Cry by inducing conformational change events that enable the Cry protein to interact with its signaling partner proteins (Figure 4). The reduced, activated form of the Cry spontaneously reoxidizes in the presence of molecular oxygen to restore the inactive form. Intriguingly, this step is accompanied by the formation of reactive oxygen species (e.g., H_2_O_2_), which can also, independently, have signaling functions [190]. A summary of the plant cryptochrome photocycle is found in Figure 4. 

#### 2.4.1. Magnetic Fields and Cryptochromes

One of the most intriguing features of cryptochrome signaling is their apparent responsivity to relatively weak magnetic fields in the range of 0–1 mT. The idea that cryptochromes may function as actual magnetosensors was first suggested for orientation in migratory birds, which orient according to the direction of the earth’s magnetic field [192]. In birds, magnetic field orientation occurs in response to the inclination of the magnetic field rather than to the north–south direction, and therefore occurs via a chemical mechanism rather than through ferromagnetic-based mechanisms known in bacterial magnetotaxis, for example. Avian orientation can moreover be disrupted by oscillating (radiofrequency) fields in the MHz range, indicative of a quantum physical mechanism. Orientation furthermore requires short wavelength light, suggesting a photoreceptor-based mechanism. The Cry1a cryptochrome was suggested as the magnetosensor specifically because it is suitably localized in the retina and is responsive to the same wavelengths as are effective in avian orientation. Furthermore, Cry1a is organized into stacks in the membrane so as to be capable of responding to an oriented signal in the bird; and it undergoes suitable redox chemistry to form possible magnetosensitive radical pair intermediates capable of responding to weak *B* (see below) [193]. 

Further evidence in support of this idea has been observations of magnetic field sensitivity demonstrated for cryptochrome responses in plants [189,194], Drosophila [195], and mammalian and human cell cultures [196] as well as in murine neuronal tissues [197], among many other examples. Because this magnetic field sensitivity occurs across phylogenetic lines in cryptochromes of many divergent origins, magnetic field sensitivity is evidently a fundamental property of the ancestral photolyase enzymes, which, in the course of evolution, has been utilized (or not) to receive magnetic field directional information from the environment. In this context, it should be mentioned that Cry4 cryptochrome, which has been suggested as a possible avian magnetosensor, is in fact not a suitable candidate. Apart from being inappropriately localized [198], the Cry4 cryptochromes have a very slow photocycle and are more similar to photolyases. For example, in the chicken (a bird that shows behavioral magnetic field orientation), the Cry4 reoxidation has a half-life of several hours, eliminating a possible role as a magnetic field sensor through formation of Trp^o^/FADH^o^ radical pairs [199]. Thus, great care should be used in assigning biological roles to specific cryptochromes.

#### 2.4.2. Radical Pair Mechanism

To explain the magnetic field response characteristics of cryptochromes, quantum physical principles have been invoked for the Radical Pair mechanism of biological magnetosensing [200]. In simplified form, this mechanism invokes the effects of applied magnetic fields on excited state intermediates in biochemical reactions that form specific spin-correlated radical pair intermediates. In this mechanism, a biochemical reaction is triggered by light or some other energy source in the cell. Two suitably positioned atoms with unpaired electrons (the so-called spin correlated ‘radical pair’) are generated with electrons in opposing spins (singlet state). The singlet state is converted spontaneously to the parallel spin state (the so-called triplet state). For this interconversion, the effect of the magnetic field is to specifically modulate the rate of singlet/triplet interconversion. Because reaction products are generally preferably formed from the triplet state, the net effect of the magnetic field will be to modulate the rate of product formation and thereby the actual reaction rate constant of a biochemical reaction. The radical pair mechanism has been supported experimentally by demonstrations using model organic compounds and in isolated proteins [200,201]. Suggested alternative mechanisms for Cry-dependent magnetosensing involving magnetite derivatives (e.g., MagR) cannot explain either the inclination compass characteristics in birds or the effect of RF fields on magnetic sensitivity [192].

In the case of cryptochromes, there are two possible steps in the photocycle where radical pairs can be formed. The first one occurs during the forward flavin photoreduction reactions (Trp^o^/FADH^o^ radical pair in the case of plant cryptochromes—Figure 4, step ‘A’). This reaction has been extensively studied in a variety of cryptochromes and photolyases and shown to be sensitive to elevated (1–10 mT) applied magnetic fields in vitro [200]. It was therefore concluded in literally dozens of publications, including many of very high impact, that the trp/flavin radical pair is involved in cry-dependent magnetoreception. However, this notion is conclusively debunked by behavioral experiments with birds, showing they can orient to the magnetic field in green light. They are also oriented when exposed to directional information in the dark (during light/dark pulse conditions) in which the Trp/flavin radical pair is not formed [192]. Similarly in plants, magnetic field effects occur by a light-independent mechanism [193,202] in which Trp^o^/FADH^o^ radical pair is not formed.

A second step at which radical pairs can be formed in cryptochromes occurs during a flavin reoxidation reaction, in which the receptor is restored to its resting (oxidized) redox state. This is referred to as reaction step ‘B’ in Figure 4. This step results from the spontaneous reaction of reduced flavin (e.g., FADH^−^) with molecular oxygen (O_2_) to form the FADox derivative (Figure 4). Although light is needed to generate the reduced flavin redox form of cryptochrome, the magnetic field effect only occurs during the reoxidation step and in the absence of light [192,193,202]. Therefore, the flavin reoxidation reaction of cryptochrome is the actual step at which biological magnetosensing must occur. 

The exact identity of possible intermediates formed at this reaction step is not fully characterized, but likely involves flavin (FADH^o^) and superoxide (O_2_
^o^-) as well as additional scavenging radicals [190,191]. What is additionally interesting about the flavin reoxidation step is that, at each iteration of the cryptochrome photocycle, two molecules of hydrogen peroxide (H_2_O_2_) are formed as byproducts. Since such so-called reactive oxygen species or ‘ROS’ have many biological effects, this reaction represents an additional way by which cryptochromes may modulate biological activity in response to *B* [203,204]. Several theoretical studies have suggested possible mechanisms for the magnetic field effects on this reaction step involving the Radical Pair Mechanism, although definitive evidence requires identification of the radicals that are formed [191]. 

#### 2.4.3. Light-Independent Magnetosensing in Cryptochromes

Intriguingly, magnetic field effects that require the presence of mammalian cryptochromes have been documented in mouse and human cell cultures [196,197]. Placing cell cultures in either lower (0–2 µT) or higher (500 µT) magnetic field strengths causes a transient increase in the production of cellular ROS. These effects are light-independent, and likely involve cellular redox reactions and flavin reoxidation. Because hydrogen peroxide (H_2_O_2_) is a byproduct of these reactions, it releases an oxidative burst in the cells that can have therapeutic effects by stimulating cellular stress response and repair pathways. Indeed, the application of *B* has been empirically used in medicine for over 50 years as a means of treatment of various diseases by means of exposure to pulsed *B* via so-called ‘Pulsed Electromagnetic Field Therapy’. Exposure to pulsed electromagnetic fields (PEMF) induces a similar increase in transient ROS in cell cultures [196], as in the case of static *B*; these magnetic field effects also appear to occur via the Radical Pair Mechanism ([205] ‘A’).

Recent evidence shows that even the Drosophila cryptochromes can mediate responses to magnetic fields independently of light, and that magnetic field sensitivity does not even require that the Cry protein binds to flavin [206]. When the Drosophila Cry C-terminal domain is expressed in flies independent of the flavin-binding N-terminal domain (see domain structure in Figure 3), the construct is still biologically active and mediates magnetic sensitivity in flies. A suggested mechanism for these intriguing effects has been that the truncated Drosophila Cry C-terminal domain may interact with other cellular flavoproteins, contributing in this way toward their signaling function without itself being directly responsive to *B*. Although the precise mechanisms involved are unknown, it is clear that if a radical pair reaction is underlying Cry magnetosensing, then it must occur independent of light, and likely involve other flavoenzyme redox reactions [206]. 

Taken together, the evidence now leads to a more generalized view of magnetosensing in living organisms, in which multiple redox-active flavoproteins and indeed mitochondrial enzymes could be modulated as a function of external *B*. Indeed, direct proof for a radical-pair-based mechanism underlying these effects has been obtained through microscopy of autofluorescence changes as a function of *B* in living HELA cells, due to the formation and electron spin-selective recombination of spin-correlated radical pairs [207]. Another example that supports the radical-pair mechanism in the spin selectivity of ROS formation in cell cultures exposed to near-null magnetic fields is found in [208]. Thus, the radical pair spin dynamics are governed by internal magnetic field interactions.

The development of single-molecule techniques such as MFM offers exciting prospective avenues in the interrogation of the magnetic response apparent in biological systems. The potential anisotropic effects applied across avian retina sections or Cry organized into crystallized arrays in vitro [208] or the existing oriented arrays in the case of avian putative magnetosensor Cry1a [192] could be accurately assessed by MFM. 

### 2.5. Biomolecules with Prospective Applications in Quantum Technologies

Biomolecules are becoming increasingly important in the field of quantum technologies. In this framework, molecules that are capable of emitting and absorbing light like the case of chromophores can be used as quantum bits [209]. The changes in the energy levels caused by photon emission and absorption render the different states of a molecular qubit [210]. For example, copper-dopped dots enhance the de-excitation electron processes as a prerequisite for single-photon sources for quantum information [211]. However, this review is focused on those biomolecules that can be exploited in quantum technologies based on their intrinsic magnetic properties. Biomolecules that present a magnetic momentum that can be modified under *B* are excellent alternatives. Ferritin, metalloenzymes, or porphyrins can work as molecular scaffolds to build the next generation of quantum technologies.

Ferritin is a ubiquitous protein found in almost all living organisms. The role of ferritin is to uptake and store iron from the media through the shell structure composed of the 24 subunits that conform to this protein [212]. Apoferritin and holoferritin terms refer to the empty protein shell and the protein with iron stored in the core (Figure 5a), respectively. Ferritin is a versatile iron nanocarrier that has received attention not only in the development of quantum technologies, but also in food nutrition, medicine [213], or as a prognosis marker in cardiovascular diseases [214]. Holoferritin is considered an affordable 3D quantum dot material that exhibits high magnetic susceptibility anisotropy at cryogenic temperatures [215]. Ferritin can contain between 3000 and 4500 iron atoms per single protein depending on their natural source [216]. Holoferritin acts as a highly disorder MNP whose net magnetic moment results from the superposition of the magnetization vectors of the different crystalline domains that make up the inner core. Interestingly, holoferritin can be trapped in self-aligned nanogaps [217], which is the preliminary step to devise more complex depositions on specific resonator locations. By this immobilization, the holoferritin molecules can detect changes in *B* with high precision. Recently, an increase in T_1_ magnetic relaxation times with the iron content inside the protein was reported [218]. This observation is due to the abrupt change in the uncompensated number of spins when the iron is accumulated inside the ferritin core. Large T_1_ relaxation times are an essential point to consider for implementing quantum error correction and fault-tolerant protocols in order to maintain quantum coherence for extended periods of time independent of the number of quantum operations. 

Metalloenzymes contain one or more metal ions in their active site, which confer their catalytic performance [219]. The metal ion stabilizes the structure of the metalloenzyme and can increase the nucleophilicity and acidity of the substrates. The most common metal ions found in metalloenzymes are iron, copper, zinc, or manganese. Metalloenzymes take part in a multitude of biological processes, including photosynthesis, and DNA repair [220], or as a target for therapeutic interventions [221]. Moreover, metalloenzymes can be artificially engineered to improve their performance [222]. The biomolecular electronic spin orientation allows us to encode the quantum bit (also named qubit) states. Layers of metalloenzymes can render bidimensional (2D) qubits. There exist numerous advantages of 2D qubits over 3D qubits: (i) 2D qubits display faster gate operations due to the close proximity of these qubits within bidimensional lattices, enabling strong coupling between the qubits in comparison to 3D qubits [223]. (ii) The coherence times of 2D qubits are generally larger based on their simpler geometries. This fact enables more complex quantum algorithms to be carried out, which request long coherence times to preserve the quantum state coherence [224]. (iii) 2D qubits contain less matter than 3D qubits, which makes them effortlessly integrate into semiconductor devices. (iv) The scalability of 2D qubits is more affordable since they are arranged in planar arrays [225]. The simple manipulation a of high number of qubits is requested for building large-scale quantum computers. For all these reasons, metalloenzymes are considered promising 2D qubits. The development of multi-functional chip devices was recently reported for recombinant azurin variants [226]. The integration of the four azurin variants with metallocenters of cobalt, nickel, iron, and manganese enables the fabrication of 4-bit memory devices. Other candidates to be converted in 2D qubits are superparamagnetic myoglobins in native and aqua-met forms (electronic structures of S = ½ and S = 5/2, respectively), where the iron metal is bonded in the heme group with octahedral and distorted trigonal bipyramidal coordinations (Figure 5b) [227]. 

**Figure 5 nanomaterials-13-02585-f005:**
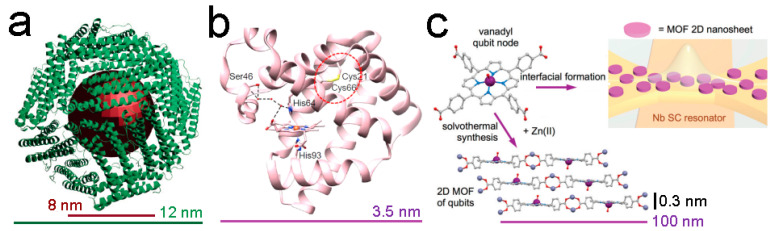
(**a**) Schematic representation of ferritin, based on the protein database (PDB) of horse-spleen apoferritin (PDB ID: 2W0O). Reprinted with permission from [217]. Copyright 2022, MDPI. (**b**) Crystal structure of variant myoglobin (PDB ID: 5ZEO). The hydrogen bonds involving the distal His64 and Ser46 residues with the heme group are shown by black dashed lines. Reprinted with permission from [228]. Copyright 2022, American Chemical Society. (**c**) Bulk and interfacial formation of the 2D MOF [{VO(TCPP)}-Zn_2_(H_2_O_2_)]_∞_. Reprinted with permission from [229]. Copyright 2020, Royal Society of Chemistry.

Finally, porphyrins are a group of large organic compounds with a heterocyclic structure formed by four pyrrole rings linked together by methine bridges [230]. Porphyrins are considered to be a subgroup of metalloenzymes since porphyrins also bind metal ions as part of their active sites. Well-defined porphyrin structures with controllable size and morphology exhibit great potential as electrochemical sensors [231], mediators of photocatalytic reactions [232], or chelating agents for molecular imaging [233]. Porphyrins treasure the property to create spontaneous metal–organic frameworks (MOFs) when a metal ion is present in the media [234]. Porphyrin MOFs can be arranged in 2D layers by targeting molecules containing a pair of weakly coupled metal coordinates or through the dinuclear complexes of metal anisotropic ions. Cu(II) phthalocyanines can be assembled through diamagnetic Zn(II) nodes creating nanosheets with optimal spin-lattice relaxation values [235]. Zn(II) can also bind forming vanadyl 5,10,15,20-tetrakis(4-carboxyphenyl)-porphyrin [{VO(TCPP)}Zn_2_H_2_O)_2_]_∞_ systems (Figure 5c) [229]. The average dimensions of the porphyrin nanosheets formed are 100 nm and 0.3 nm in diameter and height, respectively. Vanadyl porphyrins have superior spin dynamics than the analogous aforementioned copper porphyrins, which makes them interesting for coherent quantum manipulations. Another example is the case of dithiocatecholate complexes with vanadium (IV) and copper (II): [V(C_6_H_4_S_2_)_3_]^2−^ and [Cu(C_6_H_4_S_2_)_2_]^2−^ [236]. The covalency arranged in these porphyrins results in the observation of spin coherences at higher temperatures, which offers potential in manufacturing qubits in environmental conditions necessary for quantum sensing biology applications. The last porphyrin model discussed is the coordination of lanthanides (as gadolinium (III)) with 1,3 diketone and 2,6-diacetylpyridine [237] where the large separation between the magnetic quantum numbers related to the first excited and ground states encodes a two-level spin system. Moreover, the large dimensionless magnetic moment (also known as g-factor) makes the qubit computational basis enormously polarizable, providing more optimal initialization capacities. To sum up, Figure 6 summarizes all the biosystems discussed in this section.

## 3. Working Principles of MFM

MFM is a special AFM operational mode, which uses a flexible cantilever with a tip consisting of a magnetic material [100,238]. The AFM tip geometry and material vary depending on the final experimental purpose, and the fundamentals of the AFM probe selection rely on the measured physico-chemical properties [239]. The cantilever bends when touching the sample surface following the linear-Hookean force law [240]:(1)F=−kΔz
where *F* is the sensed force, *k* corresponds to the cantilever spring constant, and Δ*z* refers to the cantilever deflection, which is the direction perpendicular to the scanning plane and sample. Thermal tune [241] and Sader’s [242] methods are used to determine the spring constant of soft (*k* < 1 N/m) and stiff cantilevers (*k* > 1 N/m), respectively. Additionally, the spring constant can be alternatively estimated by other approaches like vibrometry [243].

The cantilever can be described as a simple harmonic oscillator and, excluding damping [244], its motion is described by the equation:(2)$$z\left(t\right)=A\;cos\;\left(\omega_{0}t+\varphi \right)$$
where *z*(*t*) is the precise position at a certain time, *A* is the constant oscillation amplitude, whereas *ω*_0_ and *φ* symbolize the natural frequency and the phase respecting the start point of the sinusoidal motion, respectively. Two scanner configurations exist to raster a certain surface area, named the sample scanner and tip scanner. In the first option, the sample of interest is attached on a nanoflat surface mounted on a piezoelectric scanner, whereas the scanner is coupled into the AFM tip, in the second alternative. The scanner is made of ceramic materials that enable the motion of the sample surface over the AFM probe (sample scanner) or the opposite schema (tip scanner) by applying external voltages. Then, a laser beam is reflected by the external side of the cantilever probe toward a photodetector sensing device. As an alternative to the laser, the cantilever can be read out with a piezoresistive element on the cantilever chip. This approach could be promising to prevent the potential sample damage by the laser beam. The feedback loop maintains a constant interaction force between the AFM tip and the external sample-surface-fixing-defined *A*_0_ or *ω*_0_ values. Proportional and integral derivative (PID) controllers deliver a fast feedback response according to the signal recorded by the AFM setup. The schematic representation of the AFM main components is depicted in Figure 7a.

The previous descriptions are the fundaments of the contact and dynamic modes [245]. For soft biological samples, it is advisable to avoid direct contact with the tip, hence oscillating methods are commonly used. The stationary oscillation of the cantilever at frequency *w* driven by the piezoelectric actuator is modified if a gradient of the interaction force *F_ts_* between the tip and the sample is present:(3)$$A_{0}kcos\;\left(\omega t\right)=mz^{\prime \prime }\left(t\right)+\frac{m\omega_{0}}{Q}z^{\prime }\left(t\right)+\left\{\left[k-\frac{{dF}_{ts}}{dz}\right]z(t)\right\}$$

*A*_0_ is the amplitude of the cantilever, and *m* and *Q* are the mass and the quality factor of the cantilever, respectively. Thus, the phase (*ϕ*) between the free-drive and excited cantilever oscillation is described by the following expression [246]: (4)$$\phi =arctan\;\left(\frac{m\omega \omega_{0}}{Q\left(k+\frac{dF_{ts}}{dz}-m\omega^{2}\right)}\right)\;\;$$

And the force gradient produces a phase shift given by [246]: (5)$$∆\phi =-\frac{Q}{k}\frac{dF_{ts}}{dz}$$

Notice that the gradient of the force is the magnitude measurable by the oscillating method, and *ϕ* can be influenced by modifications of *Q* and *k* during the measurement. *F_ts_* encompass the van der Waals and electrostatic forces in addition to the magnetic force. The resonance frequency also shifts with respect to *ω*_0_ [246]: (6)$$∆\omega \approx -\frac{\omega_{0}}{2k}\frac{dF_{ts}}{dz}$$

Measuring the variables included in Equations (5) and (6) has motivated numerous methods because *Q*, *A*_0_, and *ω*_0_ change as the tip-samples distance decreases.

The van der Waals force is short-ranged but is more potent than electrostatic and magnetic forces. Thus, it is essential to position the AFM probe at least 30–50 nm above the topography scan (see Section 4 for more details) to account for the long-range magnetic forces [247]. The magnetic force between a tip, which in the dipole approximation, has a fixed magnetic moment *m_tip_*, and the sample is due to the gradient of the stray field *B_stray_* generated by surface and volume magnetic charges [248]:(7)$$\vec{F}={\vec{m}}_{tip}\frac{\partial {\vec{B}}_{stray}}{\partial \vec{z}}$$
where the cantilever displacement is normal to the sample. In heterogeneous samples, the electrostatic force is present due to the varying electronic behavior. Numerous techniques have been suggested to eliminate electrostatic artifacts, which will be briefly outlined in Section 4.3. According to Equation (7), the force and its gradient can either be attractive or repulsive, depending on the orientation relative to *m_tip_* and *B_stray_*. Consequently, there will be a phase shift toward lower and higher excitation frequencies, respectively (see Figure 7b). An interesting result of this method is the simultaneous recording of the topography and phase channels (Figure 7c) that allows the magnetic response with the morphology of the observed features to be correlated. The contrast generated in the MFM channel is due to the phase shift generated by the tip-sample magnetic interaction. Parallel and anti-parallel orientations of the tip-sample magnetic moments are derived from positive and negative phase shifts, respectively (Figure 7d). Finally, another relevant aspect to consider is the cantilever spring constant. Cantilevers with low *k* render more force sensitivity but are less stable during data acquisition. On the other hand, stiff cantilevers render excellent stability during scanning, but the force detection is poor. There exists a trade-off between sensitivity and stability.

## 4. MFM Operational Modes

The methods used to reveal the magnetic features of the samples with the highest resolution rely on the use of dynamic modes. In principle, static mode could be used in the second pass of the tip over the surface as well. The AFM cantilever is lifted to a certain height to prevent detrimental short-range interactions in this second pass. For that, the average topography of the entire scanned region is set as lift height (dashed line, Figure 8a). In this case, the cantilever deflection provides information about the force (constant force mode Figure 8a) rather than its gradient. However, the sensitivity of this method is much lower compared to dynamic modes and is rarely used to obtain magnetic images. In the field of AFM, new operational methods involve analyzing the interaction with the sample at the first and second tip resonance frequencies. This advancement has been helpful in the study of thin films and nanostructured materials, and could also be applied to biological systems. Below, the most used methods in this area will be briefly described.

### 4.1. Lift Mode

One of the most extended methods to determine the magnetic structure consists of a double scan over the sample. The first pass determines the topography of the sample and the second one is carried out at a constant distance over the profile determined during the first scan (Figure 8b). That so-called lift height of the tip is selected to maximize the magnetic signal without topography interference due to the short-range distance van der Waals forces at the same time. Thus, the lift height must be individually adjusted for each tested sample prior to starting the data acquisition. This step favors stable operation and accurate data recording depending on the main scan parameters. This second therefore records the modifications of Δ*ϕ* or Δ*ω* caused by the magnetic force (see Equations (5) and (6)). Lift mode can also be employed under a bimodal configuration where the cantilever is excited at its first two harmonics [249]. Bimodal AFM can provide quantitative information on the magnetic moment and magnetic field created by a magnetized sample. Nevertheless, the main disadvantage to working in dual-pass for those magnetic features with nanometer size is the difficulty in separating the magnetic signal respecting the topography-induced forces due to the same extension decay for both contributions. Recently, some developments have been devoted to overcoming this limitation like working in single-pass dual mode [250,251]. However, this novel technology is extremely sensitive to external factors. For this reason, the reported measurements are carried out in vacuum conditions. 

#### 4.1.1. Amplitude Modulation (AM)

This method is appropriate if the sample is rough or unexplored and the quality factor *Q* is large. The control parameter to perform the scan is the amplitude of the tip at the resonance frequency obtained far away from the surface. 

#### 4.1.2. Frequency Modulation (FM)

Further refinement is the determination of w_0_ instantaneously since it is directly related to the force gradient while the phase is in the AM mode mix *Q* and *ω* (compare Equations (5) and (6)). This method is achieved by forcing the phase between the oscillation of the tip and the external source to be 90 degrees. However, more sophisticated electronics are required with phase lock loop modules.

### 4.2. Constant Height Mode

The continuous estimation of the tip surface distance is mandatory since a small drift can modify the distance between the sample and the tip in the absence of feedback. However, if the drift is small and the surface topography is very flat or well determined, other strategies can be used. Mainly, the idea is to set the scan parameters with a recorded surface or the average value, and evaluate the significant *A*, *ω*_0_, and *Q* by using either the AM or FM techniques. 

### 4.3. Electrostatic and Tip Artifacts 

Similar to magnetic forces, electrostatic forces act on long distance, and can produce artifacts in the MFM images. Thus, the MFM signal can be distorted if the sample displays potential variations across its external surface. The methods dedicated to the mapping of this interaction are electrostatic force microscopy (EFM) and Kelvin probe microscopy (KPFM). The latter measures the contact potential between tip and sample and can be operated in amplitude or frequency-modulated feedback loops (Figure 8c) [252]. The combined KPFM-MFM measurement allows compensation of the surface potential difference between the sample surface and the MFM tip to be conducted during scanning in real time. Theoretical frameworks have also been developed to decouple the influence of electrostatic forces [253]. This approach fosters the accuracy of the magnetic data recorded by MFM. Another strategy consists of modifying the magnetization of the tip because it will change the magnetic force but not the electrostatic contribution. Therefore, demagnetizing the tip will nullify or demote the magnetic signal, while inverting the tip magnetization will switch the magnetic contribution’s signs. In any case, they maintain unaltered electrostatic interactions. The removal of electrostatic artifacts has been shown to be of valuable interest in characterizing superparamagnetic nanoparticles [254], and this methodology could be extended to other systems of interest. The tip magnetization can modify the magnetic state of the sample and produce variations in two consecutive images taken of the same area of the sample. This is observed in soft magnetic materials and may require tips with small quantities of magnetic materials.

Other techniques can be used to study biological systems with high spatial resolution. However, they are technically more complex than the MFM methods described above. Two of the most interesting techniques, namely, magnetic resonance force microscopy (MRFM) and nitrogen-vacancy (NV) microscopy are briefly discussed:

### 4.4. Magnetic Resonance Force Microscopy (MRFM) 

This technique measures the magnetic force between nuclear spins in a sample and a magnetic tip, being suitable for non-magnetic samples. It combines magnetic resonance imaging (MRI) with MFM. MRFM works by applying a radiofrequency (RF) pulse, polarizing the nuclear sample spins. These nuclear spins and magnetic particles absorb energy to move to higher energy states when the RF pulse is tuned at their resonance frequencies. Then, the AFM tip detects the mechanical transitions induced by the nuclear spins, and tridimensional images of the magnetic spatial distributions can be achieved after their correlation with the positioning of the AFM tip (Figure 8d) [255]. The magnetic moment sensitivity (µ) of this technique is given by: (8)$$µ=\frac{1}{G}\sqrt{4\Gamma k_{B}Tb}$$
where *G* is the magnetic field gradient, Γ is defined as the total friction undergone by the cantilever oscillator, *k_B_* is the Boltzmann constant, *T* corresponds to the working temperature, and *b* is the detection bandwidth of the MRFM setup, respectively [256]. This expression pinpoints the critical factors that affect µ. The optimization of the µ parameter demands the use of small magnetic particles to induce large magnetic gradients [257] and to minimize as possible the sources of cantilever friction and the scanning temperature [258]. Additionally, it was recently reported that the sample coating with few nanometers of a thin metal like gold can significantly reduce the signal-to-noise ratio up to 20-fold [259], and the sensitivity of MRFM can be improved up to 10 µ when the RF coil is replaced by a microwave micro-strip resonator [260]. Finally, MRFM can monitor statistical spin fluctuations rendering larger polarizations, which have narrower distributions compared to thermal polarizations [261]. This enables MRFM to reach nearly 10 times greater sensitivity detections than classical EPR setups, thus displaying promising routes to enhance the sensitivity of weak magnetic signals coming from nanometer-scale volumes.

### 4.5. Nitrogen-Vacancy (NV) Microscopy

Nitrogen-vacancy centers are atomic fluorescent defects in diamond crystals where a nitrogen atom replaces a carbon atom adjacent to a lattice vacancy. This creates an unpaired electron spin associated with the nitrogen-vacancy center [262]. The advances and implementation of NV centers in single-molecule techniques are based on their unique magneto-optical properties like single-photon generation [263]; their energy levels, which can create a quantum bit at room temperature with an easily accessible energy spacing in the GHz range [264]; their spin state levels at local sample regions, which can be read out via optical signals [265], and their milli-second coherence times [266]. All these make NV centers attractive to be used as robust platforms for atomic-size quantum sensing [267]. For it, the NV center needs to be excited at 532 nm (green light), which will be coupled with the native state, reducing, thus, the fluorescence rate compared to the ground state. The resonance can be optically detected by sweeping the microwave frequency [268]. The shift in emitted fluorescence is proportional to the strength of the magnetic field at the location of the NV center in the known Zeeman splitting effect [269]. The sensitivity of NV microscopy is about nT. Pulsed measurements can significantly decrease the linewidth, increasing by a factor of approximately seven the sensitivity detection threshold, enabling the detection of single spins [270], which represents a breakthrough paradigm in the characterization of advanced quantum material systems. The remarkable progress made in the monitoring of the transition kinetics of single NV centers [271] has allowed the development of sensors for biological and quantum applications, especially the detection of cancer biomarkers in tumor tissues [272] and the study of radical-pair reactions and chiral spin selectivity materials [273], respectively. 

## 5. Magnetic Force Measurements with Commercially Available MFM Tips

Commercial MFM probes have benefited from great advances in the course of time. In the early days, the nominal tip radius of these probes was 60–70 nm due to their coating with cobalt-chromium (CoCr). Nowadays, AFM probe suppliers have been able to significantly reduce the tip apex radius up to 25–30 nm based on decreasing the CoCr coating thickness to 15–20 nm. Furthermore, commonly, these magnetic tips are coated with successive layers protecting them from detrimental oxidation processes. The reduction of the MFM tip radius dimensions leads to the perturbation of the magnetic signal coming from the scanned magnetic domains to be minimized, reducing, thus, the reversal during imaging and improving the MFM data quality [274]. The nominal magnetic moment of these commercial tips is nearly 1 × 10^−13^ emu. Decreasing the magnetic moment of the MFM tips is crucial to measuring samples with low coercivity values. Another alternative to detect materials with low coercivities is to coat the AFM tips with alloys of hard magnetic materials like iron-palladium (FePd), iron-platinum (FePd), or cobalt platinum (CoPt) [275]. Nevertheless, extremely high temperatures of almost 600 °C need to be employed, which can be derived in processing issues like side lateral reactions with the core silicon tip or unbalanced migration of the magnetic deposited alloy [276]. For these reasons, commercial manufacturers prefer the use of CoCr even if this material does not sense the same magnetic field strength in comparison with other alloys with different chemistry by attempting to reduce the AFM tip dimensions as much as possible. 

Many research works have been carried out to interrogate the magnetic response coming from biology systems or nanomaterials applied for biological applications. Table 1 depicts the most relevant MFM studies applied in this field. Some key insights can be gathered by observing the data displayed in Table 1. The most used MFM operational mode is based on lifting the MFM tip during a second pass. Lift mode is the most widespread among the users because this approach avoids the detrimental contribution of short-range electrical and van der Waals interactions and it is more user-friendly and cost-effective than other more complex techniques such as MRFM, as indicated in previous sections. Moreover, there is no guideline to select a proper lift height. Some works reported lift heights of 10 nm and others of several hundreds of nm. The magnetic scanned feature size limits this operational setting since the magnetic signal strongly decays with distance. 

We can conclude that most of the samples analyzed by MFM come from magnetic nanoparticles based on the lack of sensitivity of magnetic-coated MFM tips commercially available to measure magnetic entities with minor sizes. MFM was able to measure magnetic features with a broad range of shapes but the most common scanned morphology is globular with a lower detection limit settled at 3.8–4.5 nm of the particle diameter [288,292]. This is the inspirational reason to devote more research to designing and fabricating more sophisticated MFM tips reducing the tip apex dimension.

## 6. Development of Ultra-Sharp MFM Tips

A prerequisite for high-resolution AFM images is an ultra-sharp tip. However, in the case of MFM, it must be pointed out that the spatial resolution is a compromise between the tip apex and magnetic sensitivity, with the latter being determined by the quality but also the quantity of the magnetic material [299]. 

Conventional MFM probes are typically standard AFM tips that are coated with magnetic materials by sputtering. For a sufficient magnetic signal, a certain thickness of the functional coating is required, which inevitably increases the tip radius. Commercially available MFM probes provide typical tip radii of about 30 nm or more. Such broad tips, however, do not routinely allow high-resolution MFM as required in biological systems.

In this section, different development paths to reduce the tip radius below 30 nm are presented. Figure 9 depicts some representative MFM tips for each fabrication technique for a proper overview.

### 6.1. Advanced Coating Approaches

Coating AFM cantilevers by sputtering is a straightforward approach and can be easily scaled up to wafer level to cover the original tip with high-quality material. There are a few approaches to increase the resolution of MFM measurements while still using a coating technique. As already mentioned in Section 5, the most intuitive approach is to reduce the thickness of the coating and/or to use ultra-sharp AFM tips as a scaffold [300] (Figure 9a). This can reduce the radius down to 15 nm (e.g., SSS-QMFMR from NANOSENSORS^TM^ [301]), but at the expense of the magnetic properties [302] due to the reduced material volume, while the increased risk of delamination still remains. The second option is to sputter only one side of the tip, which effectively reduces the tip radius [303]. However, aside from the same disadvantages as for a full coating (delamination, tip wear, and sensitivity), there is also the slight lateral offset between the topography and magnetic signal due to the asymmetric coating, which needs to be considered in data analyses. 

An unconventional approach is the so-called “dual-tip” MFM probe introduced by Precner et al. [304]. The basic idea is to deposit magnetic material on one sidewall of the tip and then separate the original tip from the cantilever by focused ion beam milling. This creates two closely spaced tips, one magnetic and the other non-magnetic. Using a two-pass method, in the first pass, the non-magnetic part of the dual-tip is excited and used for topography acquisition. In the second pass, the magnetic part follows the sample topography at a user-selected lift height to measure the magnetic tip-sample interaction. The main advantage of this approach is that the magnetic tip is not in close contact with the sample (as required in the first pass), which minimizes the risk of disturbing magnetically sensitive samples [304]. In terms of lateral resolution, the non-magnetic tip can be a high-resolution standard AFM tip, providing high-resolution information at least for the topography measurement. Combining this dual-tip approach with ultra-sharp tip fabrication methods as discussed in the following could be a potential future perspective for MFM on sensitive nanoscale samples. 

### 6.2. Nanomachining by Focused Ion Beam Milling

A Focused Ion Beam (FIB) can be employed in a top-to-bottom approach to shape a tip down to nanoscale dimensions. The simplest way is to sculpt a sharp tip from a sputter-coated probe [305,306]. If necessary, a protective carbon cap can be deposited by Focused Electron Beam Induced Deposition (see also Section 6.4) to protect the coating from sputtering [307]. Furthermore, particles with exceptional magnetic properties can be placed at the tip region and finally shaped via FIB to the desired morphology. For example, micrometer-sized blocks were lifted out from magnetic multilayer films (e.g., SmCo_5_ [308]), and machined to a sharp tip apex with the ion beam. Campanella et al. showed the FIB nanomachining of a NdFeB milled into a hard magnetic MFM pillar [309] (Figure 9b). 

While this approach is excellent in terms of material quality, the process steps for lift-out/transfer and final shaping are cumbersome, time-consuming, and not scalable to industrial production. In addition, ion beam bombardment and ion implantation often degrade the magnetic properties of the thin film [310]. While all reported FIB-milled MFM probes were processed with Ga^+^ ions, recent advances in ion beam technology for novel ion sources might revitalize this avenue for MFM tip fabrication [311].

**Figure 9 nanomaterials-13-02585-f009:**
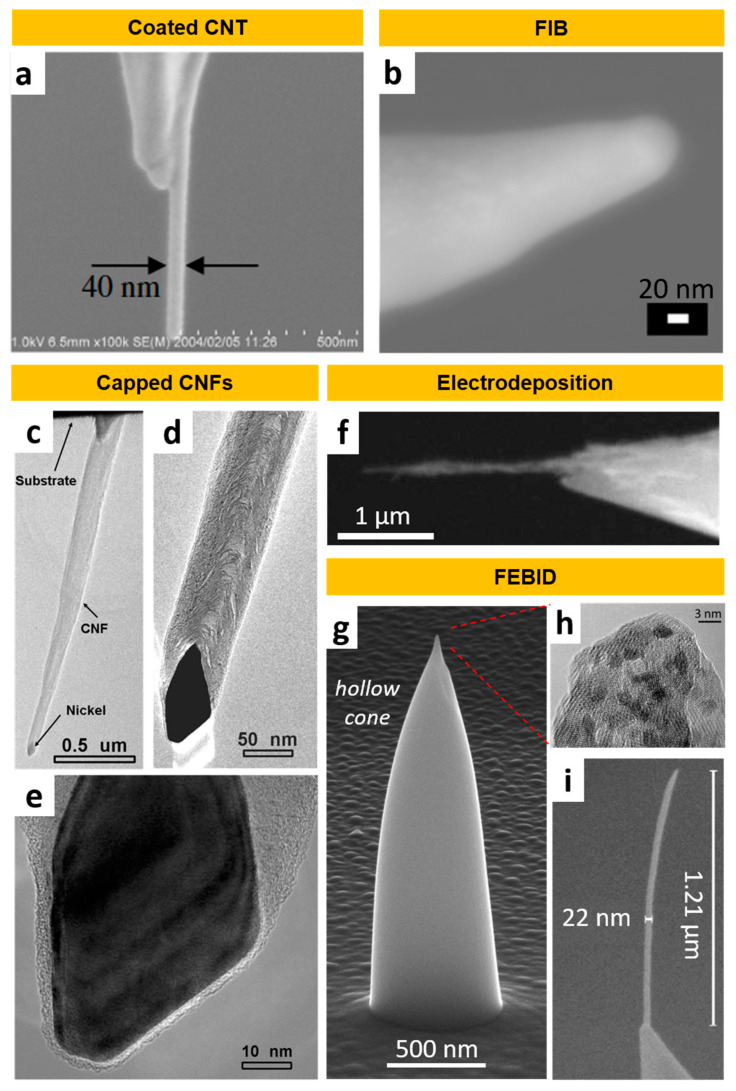
Examples of ultra-sharp MFM tips, fabricated by advanced fabrication techniques: (**a**) Carbon nanotube (CNT) after CoFe sputtering. Reprinted from [312]. Copyright 2005, IOPscience. (**b**) NdFeB needle extracted and sharpened by Focused Ion Beam (FIB) milling. Reprinted from [309]. Copyright 2011, IOPscience. (**c**–**e**) TEM images at different magnifications of Ni-capped Carbon nanofibers (CNF) grown by direct-current plasma-enhanced chemical vapor deposition. Reprinted from [313]. Copyright 2004, ACS Publications. (**f**) Ni nanowire synthesized by electrodeposition and attached to a cantilever by dielectrophoresis. Adapted from [314]. Copyright 2005, AIP Publishing. (**g**) Hollow Co_3_Fe cone deposited by Focused Electron Beam Induced Deposition (FEBID). Reprinted from [315]. Copyright 2023, MDPI. (**h**) TEM image of the tip area of a Co_3_Fe FEBID MFM tip. Reprinted from [315]. Copyright 2023, MDPI. (**i**) Extremely thin Fe FEBID pillar deposited on an AFM tip. Adapted from [316]. Copyright 2021, MDPI.

### 6.3. Carbon Nanotubes, Carbon Nanofibers, and Electrodeposited Wires

Carbon nanotubes (CNTs) are commonly used for ultrahigh-resolution AFM tips. For MFM, CNTs must be either coated [300,312,317] (Figure 9a), capped (magnetic particle at the end) [313,318] (Figure 9c–e), or filled with magnetic materials [319,320,321]. 

The common advantage of such MFM probes is the fact that the magnetic material is widely protected from oxidation by the carbon shells [321]. Wolny et al. demonstrated further advantages such as monopole-like behavior and exceptional long-term performance of Fe-filled CNTs grown by chemical vapor deposition with a ferrocene precursor [321]. 

A major challenge for CNTs, however, is the transfer to the tip area. This can be performed via micromanipulators or an electric DC field [318]. In general, those are risky, cumbersome, and time-consuming procedures and, consequently, not suitable for large-scale production [321]. Another approach is to grow a large number of CNTs directly on an AFM cantilever by CVD [322]. This eliminates the laborious mounting procedure; however, the positioning and orientation of the wires cannot be fully controlled [323]. It should also be noted that the high temperatures (several hundred °C) [318,322] involved in CVD are not compatible with all cantilever platforms.

Cui et al. used e-beam lithography to deposit Ni and Fe directly on a tipless cantilever as seeds for the growth of carbon nanofibers (CNFs) in a direct-current plasma-enhanced chemical vapor deposition reactor [313]. The cone-shaped carbon fibers terminate with the magnetic nanoparticles, as shown in Figure 9c–e. 

Magnetic nanowires (Ni, Co) can also be synthesized by electrodeposition [314] (Figure 9f). Yang et al. furthermore showed the attachment to AFM tips via dielectrophoresis, where the suspended wires were anchored to the Si pyramid under an AC field [314].

### 6.4. Focused Electron Beam-Induced Deposition 

Focused electron beam-induced deposition (FEBID) is an additive, direct-write technique based on the local decomposition of precursor molecules by a focused electron beam [324,325,326]. A gas injection system continuously delivers gaseous precursor molecules into the vacuum chamber, where they adsorb, diffuse and eventually desorb again from the surface. The interaction with the electron beam dissociates the precursor and locally immobilizes the functional material. FEBID technology offers some significant and partly unique advantages for the fabrication of ultra-sharp AFM probes: (1)Cylindrical pillars that taper to a sharp tip apex with radii of less than 10 nm [315,327] (Figure 9h).(2)Customizable pillar heights that are defined by the electron exposure conditions, allowing for tips with high aspect ratios (Figure 9i), which is beneficial for a quasi-monopole behavior [328].(3)The magnetic volume can be deposited precisely at the tip region; either on a FIB-milled or FEBID-grown plateau [329], onto an existing tip [330] (Figure 9i), or directly on tipless cantilevers [315]. While the first approach requires an additional process step, the second is a straightforward single-step process. Fabricating on flat/tipless cantilevers requires a more sophisticated FEBID-tip design [315] (Figure 9g), but simplifies the production of more advanced cantilever layouts.(4)For perpendicular alignment of the cantilever axis to the substrate plane, the technical pre-tilt in AFMs (typically about 10°) can be easily compensated [330].(5)FEBID is typically performed at room temperature [331], thus avoiding thermal stress for the cantilever.(6)Flexibility in material properties: The first attempts used the FEBID pillars as a scaffold for sputtering with magnetic materials [332,333]. The development of high-quality magnetic precursor materials for FEBID [325,334] has made this second process step unnecessary, now allowing for true direct-write, single-step fabrication of all magnetic tips [335]. Consequently, FEBID-MFM tips have no risk of delamination, while revealing 10 nm apexes. Different precursor materials have been used for FEBID-MFM probes, listed in Table 2.(7)Tip dimensions and material quality can be adjusted by the deposition conditions, such as primary electron energies and beam currents, which enable a controlled tuning of magnetic properties [327,336]. This way, FEBID-MFM tips can be adapted to the requirements of the sample and environmental conditions. For example, Jaafar et al. demonstrated exceptional MFM performance under liquid conditions [327] using Fe-based nanorods, which is highly relevant for biological samples. In addition, a range of various post-processing procedures (annealing [337], electron beam irradiation [315]) opens the door to a wide variety of MFM probes with different properties. Looking to the future, the potential of FEBID has not yet been fully exploited, considering the unrivaled possibilities of 3D nanoprinting [338] for the fabrication of advanced probe designs [315,331] (Figure 9g).

Other challenges in MFM tip fabrication via FEBID are sometimes low metal contents (depending on precursor and process parameters [325]) and low throughput, which limits FEBID to prototyping and small batch production. The advent of multi-beam instrumentation, however, contains the potential to change the situation and take advantage of the partly unique FEBID advantages. Table 2 lists the most relevant studies on the fabrication of ultra-sharp MFM tips. 

**Table 2 nanomaterials-13-02585-t002:** Ultra-sharp MFM tips reported in the literature, sorted by the size of the tip radius. The first column indicates the fabrication technique and the tip shape. The second column lists the first author of the study. The third column gives the magnetic material or, in the case of FEBID, the precursor material. Forth column gives the tip radius, or an estimation from images if the value is not reported explicitly.

Technique-Tip Type	First Author	Material/Precursor	Tip Radius	Ref.
CNT-filled	Wolny	FeC	n.a. ~25 nm	[320]
CNT-filled	Wolny	FeC	25 nm	[321]
FEBID-Pillar	Utke	Co_2_(CO)_8_	25 nm	[339]
FEBID-Pillar	Gavagnin	Fe(CO)_5_	n.a. (<20 nm)	[330]
Electrodeposition	Yang	Ni, Co	20 nm	[314]
FIB milling	Campanella	NdFeB	20 nm	[309]
CNT-coated	Kuramochi	CoFe	20 nm	[312]
CNT-capped	Arie	Ni_3_C	17 nm	[318]
CNT-coated	Deng	Ti/Co/Ti	15 nm	[319]
FIB milling	Gao	CoPt	15 nm	[306]
CNT-filled	Tanaka	Co_3_C	15 nm	[322]
CNT-coated	Choi	Co_90_Fe_10_	15 nm	[317]
FEBID-Pillar	Escalante-Quiceno	Fe_2_(CO)_9_	15 nm	[340]
FIB milling	Phillips	Co	12 nm	[305]
CNF-capped	Cui	NiC	10 nm	[313]
FEBID-Pillar	Belova	Co_2_(CO)_8_	10 nm	[329]
FEBID-Cone	Winkler, Brugger-Hatzl	HCo_3_Fe(CO)_12_	9 nm	[315]
FEBID-Pillar	Pablo-Navarro	Fe_2_(CO)_9_	8 nm	[316]
FEBID-Pillar	Jaafar	Fe_2_(CO)_9_	7 nm	[327]

## 7. Discussion and Future Perspectives

Characterization of the magnetic performance of small molecules existing in nature is a topic of paramount importance not only to gain insights into their biological role, but also due to the promising implementations in spintronics in which they can be involved. Many bulk techniques like SQUID, VSM, MOKE, EPR, or AC susceptibility can determine the ensemble average magnetic response of the tested sample, but the impossibility of mapping the scanning areas of interest does not allow local magnetic signals to be discerned. MFM is presented as a suitable alternative to conduct single-molecule studies and overcome the above-described limitations. 

Traditionally, MFM is capable of being combined with other magnetic bulk techniques such as MOKE, which enables ultrathin films of varying temperatures in vacuum conditions [341] or randomly oriented magnetic electrospun nanofibers to be studied [342]. Then, MFM measurements can be also aligned with TEM-based DPC techniques. The gathered data are complementary because MFM and TEM-based DPC achieve the out-of-plane and in-plane sample magnetization, respectively. This experimental configuration was devoted to eliciting the magnetic domain structures of crystalline spinodal alloys at the nanoscale [343]. Finally, MFM can be also coupled with other SPMs such as scanning tunneling microscopy (STM) [344] by supplying one scanner for each operational tool. STM can reveal the charge distributions based on the sample electron states at atomic resolution [345]. MFM-STM can be exploited to measure the electron transport between the nanogap junctions between two electrodes and the associated magnetic signal [346]. The main problem associated with STM is that the tip surface diffusion at environmental conditions is significant [347], which limits the applications of MFM-STM in biology. Further progress is to upgrade MFM setups with quartz tuning fork qPlus sensors [348]. The use of stiff quartz sensors optimizes the signal-to-noise ratio and improves the frequency contrast resolution up to the level of mHz [349]. This threshold is sensitive enough to detect the magnetic dipole–dipole interactions under unequal relaxation dynamics [350]. Finally, multioperational MFM probes functionalizing the tip apex with biotinylated DNA are capable of attaching a single avidin–ferritin conjugate entity and, then, interacting with modified surfaces of ferritin [351]. This approach enables the magnetic signal and the tip-sample intermolecular adhesion forces to be simultaneously acquired. Thus, MFM is shown as a multiparametric technique identifying multiple sample properties at the nanoscale. The main bottleneck of MFM measurements is the shape of the tip apex. MFM tips commercially available can detect the magnetic response of features higher than 4.0–5.0 nm. Recently, much effort has been devoted to fabricating MFM tips with reduced tip radius. We expect that these ultra-sharp MFM tips with unprecedented resolution will serve to interrogate local sample regions with a much lower number of spins. Furthermore, recent advances in 3D nanoprinting [338,352,353,354,355] using FEBID have opened new avenues for advanced MFM tip designs [282,315], such as hammerhead tips [356] or meshed-styled nano-cubes [357]. These high-resolution 3D capabilities, combined with ongoing research on new precursor materials and post-processing approaches, make FEBID a promising candidate for novel MFM probes. For all the above-described reasons, the future prospects of MFM are excellent [358,359] to deal with the current challenges, not only in the field of quantum technologies, but also for drug delivery [360], tissue regeneration [361], and wastewater treatments [362]. 

The magnetic characterization of small features independent of their source could open new gates in the design and miniaturization of superconducting resonators that can act as ultrasensitive paramagnetic resonance devices. The development of quantum technologies requires the proper attachment of magnetic features with multiple spin states on these resonator chips. The Langmuir–Blodgett (LB) technique allows homogeneous films controlling the coverage of the deposited material through the surface pressure to be created [363]. LB has successfully been exploited to transfer successive layers of porphyrins on niobium surfaces of superconducting coplanar resonators [229]. The main drawback related to LB is that it is not possible to deposit matter on specific areas of interest rather only to the entire surface introduced to the air–liquid interphase. This aspect is relevant when nanoconstrictions are specifically fabricated in the resonator chip [364]. These regions concentrate the microwave magnetic field due to the spin Hall nano-oscillator effect [365], which gives the maximum orientation and subsequent coupling of the spins with the photon magnetic field, improving, thus, the signal sensitivity required for quantum applications. Soft dip-pen nanolithography (DPN) overcomes this limitation, being able to deposit the material of interest embedded in inks under certain conditions of relative humidity and viscosity [366]. Magnetoferritins have been deposited on specific SQUID sensor regions by DPN [367]. This approach enables the alternating current magnetic susceptibility of these ferritin arrays in a wide range of temperatures and frequencies to be measured. DPN could be also employed to precisely control the deposition on the customized nanoconstrictions of other magnetic biomolecules like the aforementioned porphyrins or matter of different a nature such as metalloenzymes, or small magnetosome nanoparticles. Moreover, the creation of 2D- and 3D-lattice architectures of nanometric magnetic features can make tunable systems by out-of-plane ligation by scanning probes [368]. 

Progress toward these quantum targets requires coordinated cooperation between disciplines and the combination of experimental and simulation methods. The characterization of the deposits on the nanoconstrictions could be studied by MFM using ultra-sharp magnetic tips to quantitatively assess their magnetic response. Ultra-sharp MFM tips have been shown to exhibit significantly more sensitivity than commercial MFM tips, which can open promising future avenues in the magnetic characterization of 2D qubit systems like arrays of metalloenzymes or porphyrins (Figure 10). Cryptochromes are also presented as suitable candidates to work as 2D qubits. The radical pair mechanism involved in cryptochromes favors their quantum coherence and entanglement [369]. This allows the manipulation of cryptochromes to create functionalized surface materials maintaining the pairwise tunable spin–spin exchange quantum dynamics [370]. Furthermore, cryptochromes enable the read-out of the quantum states of single trapped molecular ions, preserving biomolecular integrity and its quantum state itself [371]. This fact can lead to the design of harnessed methodologies to couple this kind of biomolecule with an exceptional nature on quantum sensing surfaces. For all these reasons, cryptochromes may lead the next generation of the integration of 2D qubits in quantum technology resonators. Recently, MFM measurements were coupled to field gradient mapping by controlling multiple vibration modes [372], and a microwave probe station was integrated with an MFM setup to furnish simultaneous electrical and microwave contact with an operational spintronic resonator [373]. These advances will make a fine contribution to mapping local magnetic fields of these nano-oscillators. Additionally, machine learning [374] and finite-element simulations [375] can be unfurled to better understand the magnetic signal obtained by MFM through theoretical algorithm frameworks. These developed automated strategies can significantly make easier the handling and processing of the acquired magnetic data. The technology presented in this review is expected to pave the way for the development of quantum technologies.

## Figures and Tables

**Figure 2 nanomaterials-13-02585-f002:**
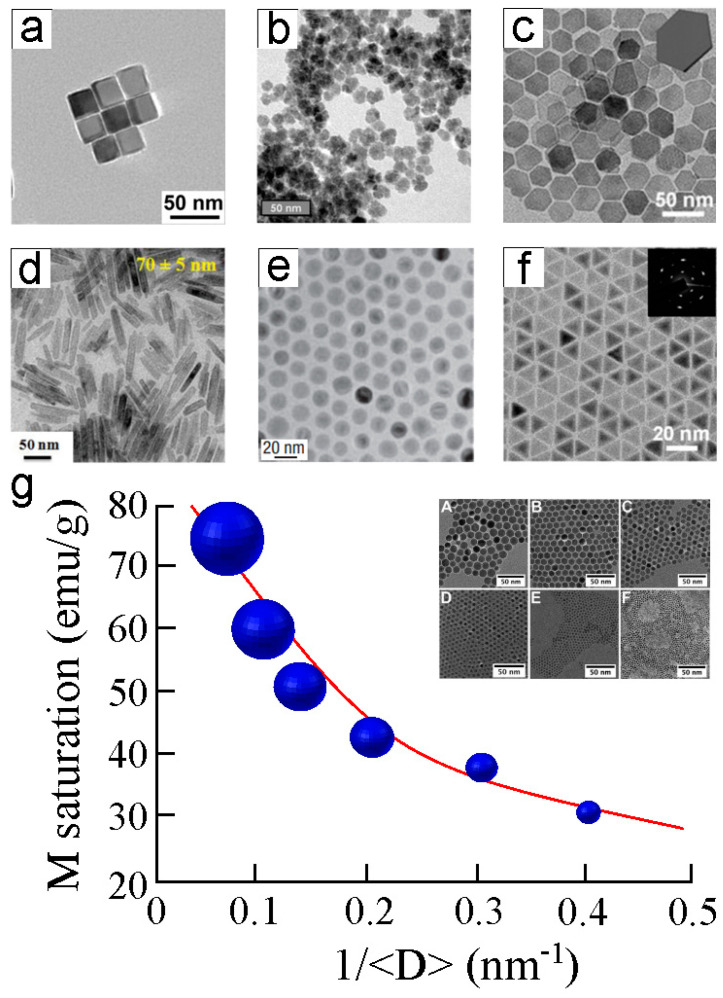
TEM images of iron oxide nanoparticles with the following morphologies: (**a**) Cubic. Reprinted with permission from [137]. Copyright 2020, American Chemical Society. (**b**) Nanoflower. Reprinted with permission from [138]. Copyright 2022, Nature. (**c**) Hexagonal. Reprinted with permission from [139]. Copyright 2015, American Chemical Society. (**d**) Rod-shape. Reprinted with permission from [140]. Copyright 2015, Royal Society of Chemistry. (**e**) Globular. Reprinted with permission from [141]. Copyright 2004, Nature. (**f**) Tetrahedron. Reprinted with permission from [139]. Copyright 2015, American Chemical Society. The scale bars are 50 nm and 20 nm for (**a**–**d**) and (**e**,**f**), respectively. (**g**) Saturation magnetization evolution as a function of iron oxide magnetic nanoparticle diameter at 5 K. The offset corresponds to the TEM images of the tested magnetic nanoparticles (MNP diameters from 14.0 nm to 2.5 nm). Inset subfigures (A–F) are TEM images of MNPs with the tested diameters. Reprinted with permission from [144]. Copyright 2011, Royal Society of Chemistry.

**Figure 3 nanomaterials-13-02585-f003:**
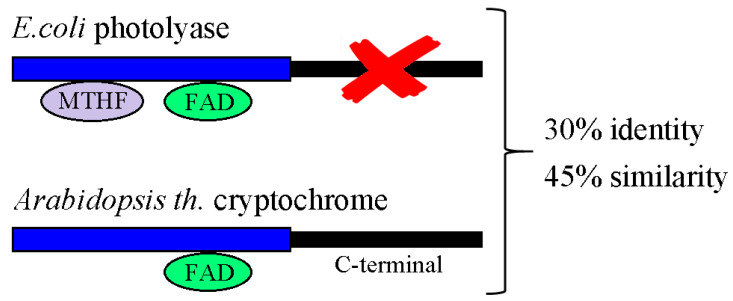
Schematic representation of the domain structure of photolyase and cryptochrome gene families. A comparison between the domain structures of *Escherichia coli* photolyase and *Arabidopsis thaliana* cry1 is shown. All classes of cryptochromes and photolyases contain the highly conserved N-terminal domain binding light-abosrbing flavin adenine dinucleotide (FAD). *E. coli* photolyase in addition binds methenyltetrahydrofolate (MTHF) antenna pigment. By contrast, the C-terminal domain is not found in photolyases and it is poorly conserved displaying variable lengths even among cryptochromes of the same species (e.g., Atcry1 and Atcry2).

**Figure 4 nanomaterials-13-02585-f004:**
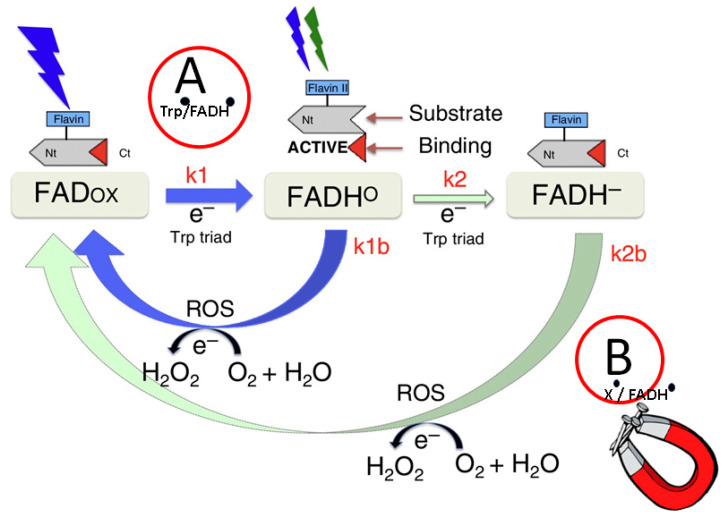
The *Arabidopsis* cryptochrome photocycle. This figure represents a composite consistent with published data. In the dark, cryptochromes are in the inactive state (C-terminal domain folded against the protein, flavin in the oxidized redox state). Upon illumination with blue light (wavelengths below 500 nm), flavin undergoes photoreduction to the FADH° redox form (rate constant k1) by forward electron transfer via the Trp triad pathway [177]. This event triggers conformational change and unfolding of the C-terminal domain to give the activated form of the receptor, which is thereby accessible for signaling partner binding. Subsequent illumination of FADH° with an additional photon of either blue or green light (wavelengths below 600 nm) can induce further reduction to the (FADH^−^) inactive redox form (k2), although at much lower efficiency than k1. Reoxidation to the resting (FADox) state from FADH° occurs spontaneously in the presence of molecular oxygen, with a rate constant (k1b) of several minutes, and is accompanied by the formation of ROS and H_2_O_2_. More rapid reoxidation occurs from the FADH^−^ redox form to FADox by an alternate pathway involving formation of transient oxygen and flavin radical intermediates [190,191]. Changes in rate constants k2b and k1b would explain the change in biological activity under applied magnetic field conditions. Reprinted with permission from [189]. Copyright 2016, Elsevier.

**Figure 6 nanomaterials-13-02585-f006:**
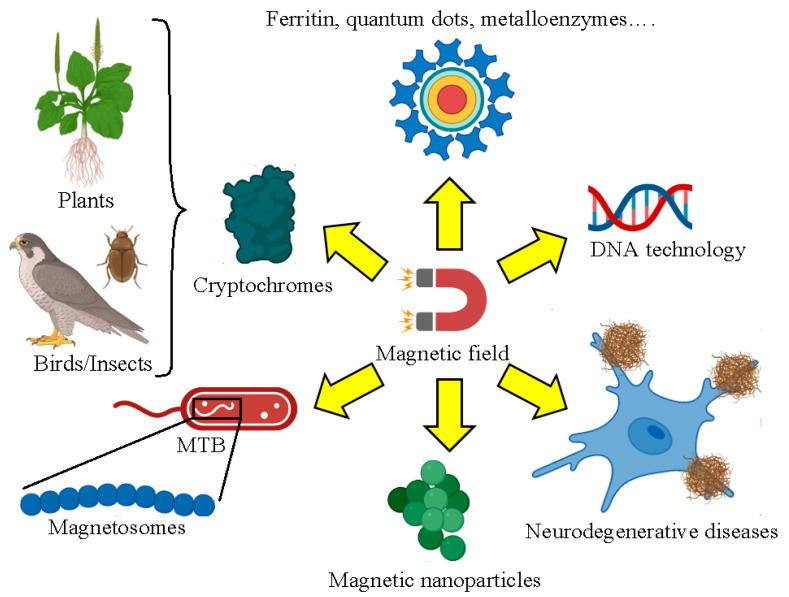
Biology systems and nanomaterials involved in biology applications affected by external magnetic fields. Images were created using BioRender.com (accessed on 13 March 2023).

**Figure 7 nanomaterials-13-02585-f007:**
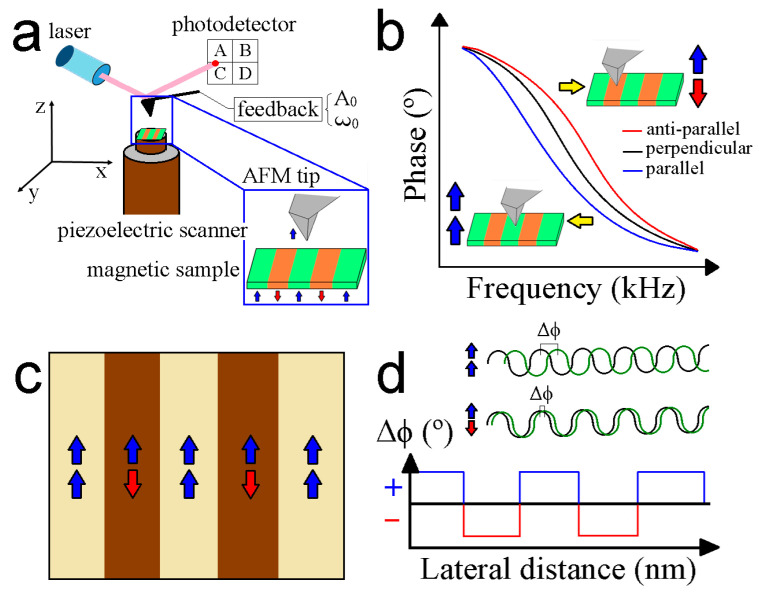
(**a**) Schematic representation of the main components of a typical AFM setup. The laser beam is reflected at the cantilever top surface to a photodetector, which records the cantilever deflection. The cantilever is excited by the feedback chosen depending on the MFM operational mode used. The piezoelectric scanner enables the high positioning precision of the mounted sample with respect to the AFM tip. The zoom inset represents the magnetic moments of the AFM tip and a multi-domain sample. (**b**) Excitation frequency shifts according to the orientation between the magnetic moments of the AFM tip and the scanned sample. Positive frequency shifts are observed when the magnetic moments are placed in anti-parallel (red line) orientation caused by repulsive magnetic forces. Parallel (blue line) orientation of the tip-sample magnetic moments and the generated attractive magnetic forces induce negative excitation frequency shifts. No changes in the frequency are reported if perpendicular (black line) tip-sample magnetic orientation moments are displayed. (**c**) MFM channel of the scanned substrate surface where the parallel and anti-parallel tip-sample magnetic orientations correspond to brighter and darker setting colors, respectively. (**d**) Phase shifts (Δ*ϕ*) originated from the magnetic tip-sample interaction. Anti-parallel orientation of the magnetic moments between the AFM tip and the scanned sample surface produces positive Δ*ϕ*, whereas the opposite effect takes place for the antagonistic parallel magnetic moment orientation.

**Figure 8 nanomaterials-13-02585-f008:**
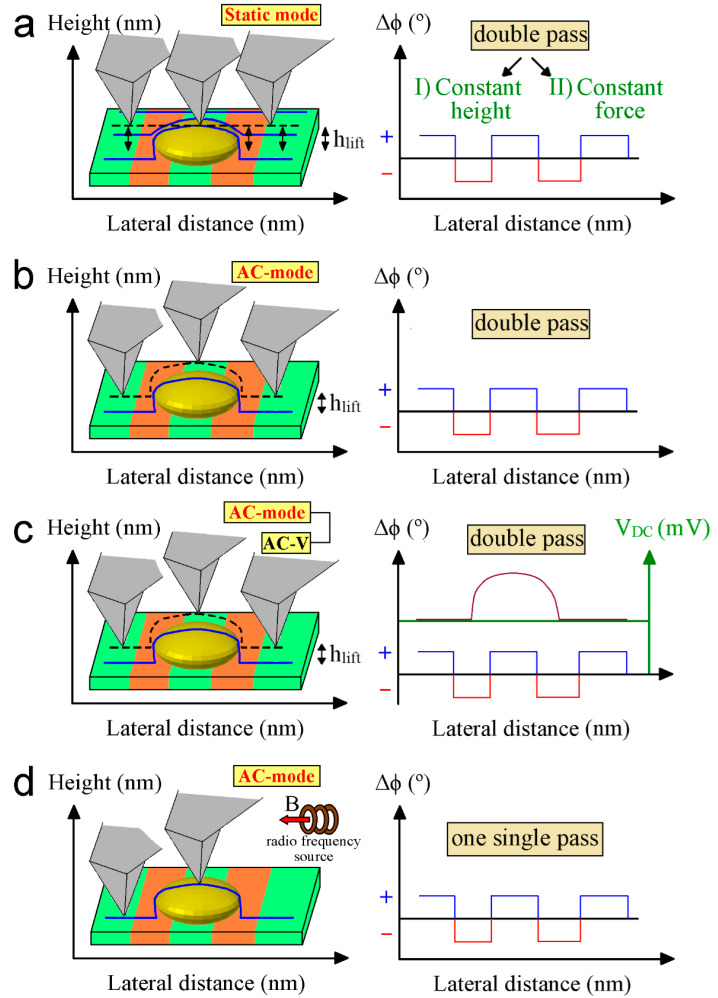
Schematic representation of all existing MFM operational modes: (**a**) Constant height and constant force modes consist of a double pass of the AFM tip close to the sample surface area in static mode. The tip-sample distance and the applied force between the tip and the external sample surface are kept constant for each mode, respectively. Left image represents the phase shift caused by the magnetic contrast (Δ*ϕ*) of the scanned sample domains. (**b**) Lift mode works with a second pass of the AFM tip above to the scanned sample surface. While in the first pass, the tip is close to this surface, during the second pass, the AFM tip is moved away a certain distance (*h_lift_*) in dynamic contact mode (AC-mode). The movement of the AFM tip in this second pass corresponds to the previously recorded topography of the scan line in the first pass. (**c**) Frequency-modulated Kelvin probe microscopy directly observes the difference in the contact potential between the AFM tip and the sample (V_DC_) by the detected probe resonant peak sidebands induced by the alternative current voltage (AC-V). The detection of Δ*ϕ* is similar to the case of lift mode. The positive and negative magnetic domains of the substrate are in green and brown, respectively. (**d**) Magnetic resonance force microscopy displays a similar configuration to the above described but the tip scans the sample surface with one single pass and setup is coupled with a microwave radio frequency source. The yellow spheroid depicts a non-magnetic feature with conductive potential.

**Figure 10 nanomaterials-13-02585-f010:**
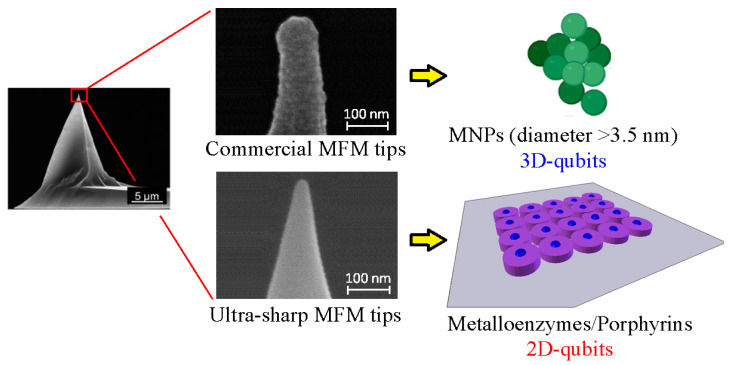
SEM field image of an MFM tip (scale bar of 5 µm). Reprinted with permission from [376]. Copyright 2010, AIP Publishing. Zoomed SEM images of the apex from a commercial and ultra-sharp MFM tips. Scale bar of 100 nm. Reprinted with permissions from [315]. Copyright 2023, MDPI. Commercial and ultra-sharp MFM tips can characterize the magnetic properties of 3D qubits and 2D qubit systems, respectively.

**Table 1 nanomaterials-13-02585-t001:** Compilation of MFM measurements carried out discriminating the biology system of interest, the MFM operational mode used, and the size of the measured magnetic features. Dimensions are assumed to be globular (indicated as “glob.”) for all the observed features with exception of the square shape of cobalt nanorings [277] and rod-shaped magnetic particles that appeared in *Magnetospirillum magnetotacticum* [278] and cobalt nanowires [277], respectively. *Magnetospirillum* is noted as *M. spirillum*. L. and H. are length and width, respectively.

Biological Sample	MFM Mode	Lift Height	Sample Height	Ref.
Magnetosomes from *M. spirillum magnetotacticum*	Lift mode	60–300 nm	~20 nm (glob.)	[278]
Magnetosomes from *M. spirillum magnetotacticum*	Lift mode	60–300 nm	~1.5 × 24 × 2000 nm (rod)	[278]
Magnetosomes from *M. spirillum gryphiswaldense*	Const. height	-	21.0 ± 2.5 nm (glob.)	[279]
Magnetosomes transfected to mesenchymal cells	Lift mode	20 nm	~12 nm (glob.)	[280]
Magnetosomes in bivalve *Thasyra* cf. *gouldi*	Lift mode	35–150	72.9 ± 28.9 (glob.)	[281]
Cobalt nanospheres	MRFM	-	~500 nm (glob.)	[282]
Cobalt nanowires	Lift mode	30 nm	~25 × 85 × 2750 nm (rod)	[277]
Cobalt nanorings	Lift mode	30 nm	1 × 0.1 µm (L., W.) (sq.)	[277]
Magnetite (Fe_3_O_4_) nanoparticles	Lift mode	50 nm	18.7 ± 3.0 (glob.)	[283]
Magnetite (Fe_3_O_4_) nanoparticles	Lift mode	10 nm	~4.8 nm (glob.)	[284]
Magnetite (Fe_3_O_4_) nanoparticles	Lift mode	10 nm	~20 nm (glob.)	[285]
Iron oxide MNPs in polymer matrix	KPFM	50 nm	~8 to 12 nm	[286]
Gadolinium nanoparticles	Const. height	150 nm	~12 nm	[105]
Fe_3_O_4_ in hydrogels	Lift mode	50 nm	34.0 ± 1.0 nm (glob.)	[287]
Iron in rodent spleen	Lif mode	30–100 nm	3.8 ± 0.2 nm (glob.)	[288]
Iron deposits in brain histological sections	Lift mode	30 nm	~5 to 8 nm (glob.)	[289]
Diphenylpicrylhydrazil (DPPH) radicals	MRFM	-	~5 to 8 µm (glob.)	[290]
Liposome membrane labeled with DPPH	MRFM	-	~5 to 15 µm (glob.)	[291]
Mitotic arrest deficient 2 (MAD2) protein	MRFM	-	~4.5 nm (glob.)	[292]
Ferritin	Lift Mode	10–50 nm	~12 nm (glob.)	[293]
Ferritin	Lift mode	30–50 nm	~12 nm (glob.)	[294]
Ferritin iron core	Lift Mode	30 nm	~5 nm (glob.)	[295]
Graphene quantum dots	Lift mode	50 nm	~6.5 nm (glob.)	[296]
Graphene functionalized with Fe_2_O_3_ particles	Const. height	-	5–10 nm	[297]
Co-FeCo dots	Cont. height	-	~25 nm (glob.)	[298]

## Data Availability

Additional data are available upon reasonable request.

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
