# Peer review of "A Review of the Current State of Magnetic Force Microscopy to Unravel the Magnetic Properties of Nanomaterials Applied in Biological Systems and Future Directions for Quantum Technologies"

_nanomaterials, 2023, doi:10.3390/nano13182585_

Round 1

Reviewer 1 Report

The Manuscript devoted to review of the applicatoin of the magnetic force microscopy for magnetic properties studying in biological systems.

This review covers both the modern techniques for magnetic studies, and possible biological objects, which are may be investigated by magnetic force microscopy technique. It is evident that such review should be in interest of many researchers, who works in the biotechnological inventions nowdays.

Nevertheless, some points should be changed in the current paper.

First, I need to note the across-the-board description of the potential systems, in which MFM can be applied. In my opinion, here the Authors admit the bias on the Cryptochromes specification compared to others objects. I would like to suggest reduce the Section 2.4.

Second, in the Introduction the Authors listed the main methods for magnetization studies, including the EPR and NMR. However, here, Mössbauer spectroscopy is not mentioned. Indeed, this method is limited in the possible isotopes, but it is a very powerful instrument for investigation. Especially, if modern synchrotron facilities are exploited. I suggest to include a short paragraph about this tool.

Author Response

13th September 2023

Dear Nanomaterials Editorial Board,

The authors would like to sincerely acknowledge to the two Reviewers for the time spent in the thoughtful assessment of the present manuscript. We enclosed the revised version of the work entitled “A review of the current state of magnetic force microscopy to unravel the magnetic properties of nanomaterials applied in biological systems and future directions for quantum technologies” (Manuscript ID nanomaterials-2609775) trying to response to the Reviewers comments. All suggestions raised by the Reviewers have been carefully considered and subsequently covered during the manuscript main body text.

Please find below the responses to the Reviewers queries attached point-by-point. The statements included in the updated review version are highlighted in red colour.                     

Reviewers` Comments to the Authors:

REVIEWER 1

The Manuscript devoted to review of the applicatoin of the magnetic force microscopy for magnetic properties studying in biological systems.

This review covers both the modern techniques for magnetic studies, and possible biological objects, which are may be investigated by magnetic force microscopy technique. It is evident that such review should be in interest of many researchers, who works in the biotechnological inventions nowdays.

Response: We sincerely acknowledge the positive comments provided by Reviewer 1 regarding our scientific manuscript.

Nevertheless, some points should be changed in the current paper.

First, I need to note the across-the-board description of the potential systems, in which MFM can be applied. In my opinion, here the Authors admit the bias on the Cryptochromes specification compared to others objects. I would like to suggest reduce the Section 2.4.

Response: We agree that the mentioned section is very elaborated and thus lengthy. However, as one central element in the review, the comprehensive description of cryptochromes justify the length according to the rest of the biological systems affected by external magnetic fields. The unpaired radical-paired electron mechanism presented in cryptochromes is singular compared with the other biological systems where the external magnetism forces play a pivotal role. We are afraid that the main message to the broad potential audience was lost in case some content of this section is erased. For these reasons, we consider keeping the current version the most appropriate alternative to aid the readers to better understand the underlying mechanisms involved in cryptochromes magnetoreception processes.

The following statement was added to emphasize the singular mechanisms involved in the cryptochrome family and place the potential readers and other target audiences about the importance of this biological systems for the examined field: “The singular mechanism that converts the cryptochromes as the unique biological receptor to sense spin chemical forces makes necessary the fully explanation of how this protein family responds under β” (lines 438-440).

In addition and taking into account the importance of this biological system on the open promising future perspectives, the following sentence was added in the latest manuscript section: “Furthermore, cryptochromes enable the read-out of the quantum states of single trapped molecular ions preserving the biomolecular integrity and its quantum state itself [371]. This fact can lead the design of harnessed methodologies to couple this kind of biomolecules with exceptional nature on quantum sensing surfaces. For all these reasons, cryptochromes may lead the next-generation of the integration of 2D-qubits in quantum technology resonators.” (lines 1217-1223).

[371] Smith, L.D.; Chowdhury, F.T.; Peasgood, I.; Dawkins, N.; Kattnig, D.R. Driven Radical Motion Enhances Cryptochrome Magnetoreception: Toward Live Quantum Sensing. J. Phys. Chem. Lett. 2022, 13, 10500-10506. https://doi.org/10.1021/acs.jpclett.2c02840.

Second, in the Introduction the Authors listed the main methods for magnetization studies, including the EPR and NMR. However, here, Mössbauer spectroscopy is not mentioned. Indeed, this method is limited in the possible isotopes, but it is a very powerful instrument for investigation. Especially, if modern synchrotron facilities are exploited. I suggest to include a short paragraph about this tool.

Response: We completely agree with the Reviewer1 suggestion. One extra bullet point was added according to the Mössbauer spectroscopy: “Mössbauer spectroscopy is a versatile technique to study the interaction of certain isotopes with their surroundings [42]. This technology enables to measure the hyperfine interactions between the nuclei and electrons with an accuracy of 14-15 magnitude orders [43]. Mössbauer spectroscopy was used to unravel the magnetic properties of nanometer-size particles [44], magnetosomes [45], or cryptochromes [46]. Mössbauer spectroscopy is a particularly powerful instrument when it is exploited in combination to synchrotron facilities [47].” (lines 112-118).

[42] Kuzmann, E.; Homonnay, Z.; Klencsár, Z.; Szalay, R. 57Fe Mössbauer Spectroscopy as a Tool for Study of Spin States and Magnetic Interactions in Inorganic Chemistry. Molecules 2021, 26, 1062. https://doi.org/10.3390/molecules26041062.

[43] Bharut-Ram, K.; Mølholdt, T.E.; Langouche, G.; Geburt, S.; Ronning, C.; Doyle, T.B.; Gunnlaugsson, H.P.; Johnston, K.; Mantovan, R.; Masenda, H.; et al. Sensitivity of 57Fe emission Mössbauer spectroscopy to Ar and C induced defects in ZnO. Hyperfine Interact. 2016, 237, 81. https://doi.org/10.1007/s10751-016-1286-5.

[44] Lin, Q.; Xu, J.; Yang, F.; Lin, J.; Yang, H.; He, Y. Magnetic and Mössbauer Spectroscopy Studies of Zinc-Substituted Cobalt Ferrites Prepared by the Sol-Gel Method. Materials 2018, 11, 1799. https://doi.org/10.3390/ma11101799.

[45] Zhu, X.; Hitchcock, A.P.; Bazylinski, D.A.; Denes, P.; Joseph, J.; Lins, U.; Marchesini, S.; Shiu, H-W.; Tyliszczak, T.; Shapiro, D.A. Measuring spectroscopy and magnetism of extracted and intracellular magnetosomes using soft X-ray ptychography. Proc. Nat. Acad. Sci. U. S. A. 2016, 113, E8219-E8227. https://doi.org/10.1073/pnas.1610260114.

[46] Bauer, T.O.; Graf, D.; Lamparter, T.; Schünermann, V. Characterization of the photolyase-like iron sulfur protein PhrB from Agrobacterium tumefaciens by Mössbauer spectroscopy. Hyperfine Interact. 2014, 226, 445-449. https://doi.org/10.1007/s10751-013-0969-4.

[47] Cini, A.; Poggini, L.; Chumakov, A.I.; Rüffer, R.; Spina, G.; Wattiaux, A.; Duttine, M.; Gonidec, M.; Fittipaldi, M.; Rosa, P.; et al. Synchrotron-based Mössbauer spectroscopy characterization of sublimated spin crossover molecules. Phys. Chem. Chem. Phys. 2020, 22, 6626-6637. https://doi.org/10.1039/c9cp04464g.

Then, the paragraph related to the TEM-based techniques was subsequently trimmed to keep the Introduction section length and thus, not to cause the non-desirable shifts of Figures and Tables.

We expect the present revised review version and the rebuttal answers settle all the comments, doubts and suggestions given by the reviewers.

The submitting authors accept responsibility for the following:

  • We have the consent from all authors to submit the manuscript and all authors accept complete responsibility for the contents of the manuscript.
  • The manuscript is not currently under consideration elsewhere and the work reported will not be submitted for publication elsewhere until a final decision has been made as to its acceptability by the Journal.
  • The manuscript is truthful original revision.

Yours sincerely,

CARLOS MARCUELLO (on behalf of all the authors)

Reviewer 2 Report

The title of the manuscript indicates it is a review of the current state of knowledge regarding the use of the MFM method for analyzing the magnetic properties of biological systems. At the same time, some of the subsections in Section 2 are too elaborate by which the unified message of the manuscript suggested by the title is lost. I propose to significantly reduce these sections and expose what is important for measurements using MFM.

Minor comments:

1) On Page 2, the authors list measurement techniques for biological materials that have magnetic properties. When measuring techniques based on magnetic resonance, only NMR and EPR are presented. Since the authors also referred to magnetic nanoparticles in the article, it would be good to add information about the FMR (ferromagnetic resonance) method.

2) On Page 2, the authors wrote in the first paragraph only about the use of gadolinium for MRI techniques.

3) On Page 7, the authors wrote "Figure 2g depicts the logarithm decay of the magnetic saturation of globular Fe2O3 MNPs." Do they mean logarithmic decreasing? It should be detailed because the dependence of saturation magnetization on the diameter shown in panel (g) is not clear.

4) The authors alternately use the Greek symbol \omega and the letter w to denote frequency, e.g., on Page 17, in the formulas and in the text. They should choose one designation.

In conclusion, the paper is very valuable but needs some reanalysis in the presentation of the MFM method and its possibilities for measuring the properties of biological materials.

Author Response

13th September 2023

Dear Nanomaterials Editorial Board,

The authors would like to sincerely acknowledge to the two Reviewers for the time spent in the thoughtful assessment of the present manuscript. We enclosed the revised version of the work entitled “A review of the current state of magnetic force microscopy to unravel the magnetic properties of nanomaterials applied in biological systems and future directions for quantum technologies” (Manuscript ID nanomaterials-2609775) trying to response to the Reviewers comments. All suggestions raised by the Reviewers have been carefully considered and subsequently covered during the manuscript main body text.

Please find below the responses to the Reviewers queries attached point-by-point. The statements included in the updated review version are highlighted in red colour.                     

Reviewers` Comments to the Authors:

REVIEWER 2

The title of the manuscript indicates it is a review of the current state of knowledge regarding the use of the MFM method for analyzing the magnetic properties of biological systems. At the same time, some of the subsections in Section 2 are too elaborate by which the unified message of the manuscript suggested by the title is lost. I propose to significantly reduce these sections and expose what is important for measurements using MFM.

Response: We thank this proposition of Reviewer2. As aforementioned described, the content of cryptochromes section (number 2.4.) is crucial to aid the potential readers to better understand the underlying mechanisms involved in the unpaired radical-paired electron mechanism in cryptochromes. This chemical reaction pathway is singular with respect to the other examined biological systems and it merits some additional explanation for those readers not involved in this field.

1) On Page 2, the authors list measurement techniques for biological materials that have magnetic properties. When measuring techniques based on magnetic resonance, only NMR and EPR are presented. Since the authors also referred to magnetic nanoparticles in the article, it would be good to add information about the FMR (ferromagnetic resonance) method.

Response: We fully agree with the suggestion made by Reviewer2. The following details were added in the bullet point that describes NMR and EPR techniques:

“Magnetic resonance techniques like nuclear magnetic resonance (NMR) [30], electron paramagnetic resonance (EPR) [31], or ferromagnetic resonance (FMR) [32] apply (…). Alternatively, FMR detects the magnetic moments of non-paramagnetic materials by applying a second microwave pulse being widely used for ferromagnetic particles [35] or magnetosomes [36]. EPR has satisfactorily accomplished (…)”. (lines 96-106).

[32] von Bardeleben, H.J.; Cantin, J.L.; Gendron, F. Ferromagnetic Resonance Spectroscopy: Basics and Applications. In Electron Paramagnetic Resonance Spectroscopy. Ed. Springer, Cham: Switzerland. 2020. Volume1. pp. 351-383. https://doi.org/10.1007/978-3-030-39668-8_12.

[35] Benguettat-El Mokhtari, I.; Schmool, D.S. Ferromagnetic Resonance in Magnetic Oxide Nanoparticules: A Short Review of Theory and Experiment. Magnetochemistry 2023, 9, 191. https://doi.org/10.3390/magnetochemistry9080191.

[36] Blattmann, T.M.; Lesniak, B.; García-Rubio, I.; Charilaou, M.; Wessels, M.; Eglinton, T.I.; Gehring, A.U. Ferromagnetic resonance of magnetite biominerals traces redox changes. Earth Planet Sci. Lett. 2020, 545, 116400. https://doi.org/10.1016/j.epsl.2020.116400.

2) On Page 2, the authors wrote in the first paragraph only about the use of gadolinium for MRI techniques.

Response: We appreciate this comment from Reviewer2. The part related to magnetic resonance imaging (MRI) starts with the potential use of gadolinium nanoparticles and its inherent safety risks. Then, iron oxide (Fe2O3) magnetic nanoparticles (MNPs) are presented as suitable alternative to overcome the above described limitations related to the use of gadolinium MNPs.

Furthermore, the following information was added to highlight the existence of MNPs with different chemistries with could be presented as excellent candidates for MRI applications: “Substitution strategies of ferrite nanoparticles by europium [129] or samarium [130] offer suitable alternatives to gadolinium MNPs”. (lines 323-324).

[129] Saeidi, H.; Mozaffari, M.; Ilbey, S.; Dutz, S.; Zahn, D.; Azimi, G.; Bock, M. Effect of Europium Substitution on the Structural, Magnetic and Relaxivity Properties of Mn-Zn Ferrite Nanoparticles: A Dual-Mode MRI Contrast-Agent Candidate. Nanomaterials 2023, 13, 331. https://doi.org/10.3390/nano13020331.

[130] Hulsure, N.R.; Inamdar, A.K.; Bandgar, S.S.; Mulik, R.N.; Shelke, S.B. Effect of samarium substitution on Mn0.5Zn0.5SmxFe2-xO4 ferrite nanoparticles. Mater. Today Proc. 2023. https://doi.org/10.1016/j.matpr.2023.04.614.

3) On Page 7, the authors wrote "Figure 2g depicts the logarithm decay of the magnetic saturation of globular Fe2O3 MNPs." Do they mean logarithmic decreasing? It should be detailed because the dependence of saturation magnetization on the diameter shown in panel (g) is not clear.

Response: This statement was modified according to the Reviewer2 suggestion: “Figure 2g depicts the logarithm decreasing of the magnetic saturation of globular Fe2O3 MNPs” (lines 355-356).

4) The authors alternately use the Greek symbol \omega and the letter w to denote frequency, e.g., on Page 17, in the formulas and in the text. They should choose one designation.

Response: Reviewer2 is right. Sections 3 and 4 (lines 730-820 and lines 821-956, respectively) were amended accordingly including the equations 2, 3 and 4 (lines 744, 766 and 770, respectively). The term of frequency was homogenized as “ω” (Greek symbol).

We expect the present revised review version and the rebuttal answers settle all the comments, doubts and suggestions given by the reviewers.

The submitting authors accept responsibility for the following:

  • We have the consent from all authors to submit the manuscript and all authors accept complete responsibility for the contents of the manuscript.
  • The manuscript is not currently under consideration elsewhere and the work reported will not be submitted for publication elsewhere until a final decision has been made as to its acceptability by the Journal.
  • The manuscript is truthful original revision.

Yours sincerely,

CARLOS MARCUELLO (on behalf of all the authors)
